# Ferroelectric freestanding hafnia membranes with metastable rhombohedral structure down to 1-nm-thick

Yufan Shen [1], Kousuke Ooe [2], Xueyou Yuan[3], Tomoaki Yamada [3,4], Shunsuke Kobayashi [2], Mitsutaka Haruta[1], Daisuke Kan [1] ✉ & Yuichi Shimakawa [1]

Two-dimensional freestanding membranes of materials, which can be transferred onto and make interfaces with any material, have attracted attention in the search for functional properties that can be utilized for next-generation nanoscale devices. We fabricated stable 1-nm-thick hafnia membranes exhibiting the metastable rhombohedral structure and out-of-plane ferroelectric polarizations as large as 13 $\mu C/cm^2$. We also found that the rhombohedral phase transforms into another metastable orthorhombic phase without the ferroelectricity deteriorating as the thickness increases. Our results reveal the key role of the rhombohedral phase in the scale-free ferroelectricity in hafnia and also provide critical insights into the formation mechanism and phase stability of the metastable hafnia. Moreover, ultrathin hafnia membranes enable heterointerfaces and devices to be fabricated from structurally dissimilar materials beyond structural constrictions in conventional film-growth techniques.

Materials in metastable phases, whose Gibbs free energies do not reside at the global minimum, exhibit chemical and physical properties distinct from those in the most stable phases and thus have great potential for novel device applications. However, stabilizing materials with metastable crystal structures requires extreme and non-equilibrium conditions[1–5], making it challenging to heterostructure them with other functional materials in devices. Recently, it has been shown that epitaxial thin films can be exfoliated from substrates by selectively dissolving sacrificial layers[6–8], namely made into freestanding membranes and stacked together with various materials. Given that materials with metastable crystal structures can be epitaxially stabilized through interfacial structural matching[9–12], fabricating and stacking freestanding membranes of such metastable materials would be a way to make heterostructures and explore their potential applications. However, while materials in equilibrium phases have been shown to form in freestanding crystalline membranes under ambient conditions and exhibit bulk-equivalent properties[13–19],

freestanding membranes with metastable structures and their physical properties have not been explored yet.

Binary oxide hafnia exhibits dielectric properties depending on its abundant structural phases, among which the most stable paraelectric monoclinic phase (Fig. 1A) has been widely studied as a CMOS-compatible high-$k$ material[20–23]. The recent discoveries of ferroelectricity in the metastable orthorhombic phase of hafnia have opened a new playground for developing ferroelectric physics and applications[2,24–26]. An important characteristic that distinguishes hafnia from conventional perovskite ferroelectrics, like barium titanate, is scale-free ferroelectricity[27–29]. In fact, recent reports have shown that hafnia ultrathin films exhibit room-temperature ferroelectricity even when the film is as thin as 1 nm[29,30]. However, structural phases and ferroelectric properties of nanometers-thick hafnia films have been reported to be susceptible to various external factors such as interfacial defects and the type of top/bottom electrode materials[31–33]. This raises concerns[34] as to whether hafnia, especially

[1]Institute for Chemical Research, Kyoto University, Uji, Kyoto, Japan. [2]Nanostructures Research Laboratory, Japan Fine Ceramics Center, Nagoya, Japan. [3]Department of Energy Engineering, Nagoya University, Nagoya, Japan. [4]MDX Research Center for Element Strategy, International Research Frontiers Initiative, Tokyo Institute of Technology, Yokohama, Japan. ✉e-mail: dkan@scl.kyoto-u.ac.jp

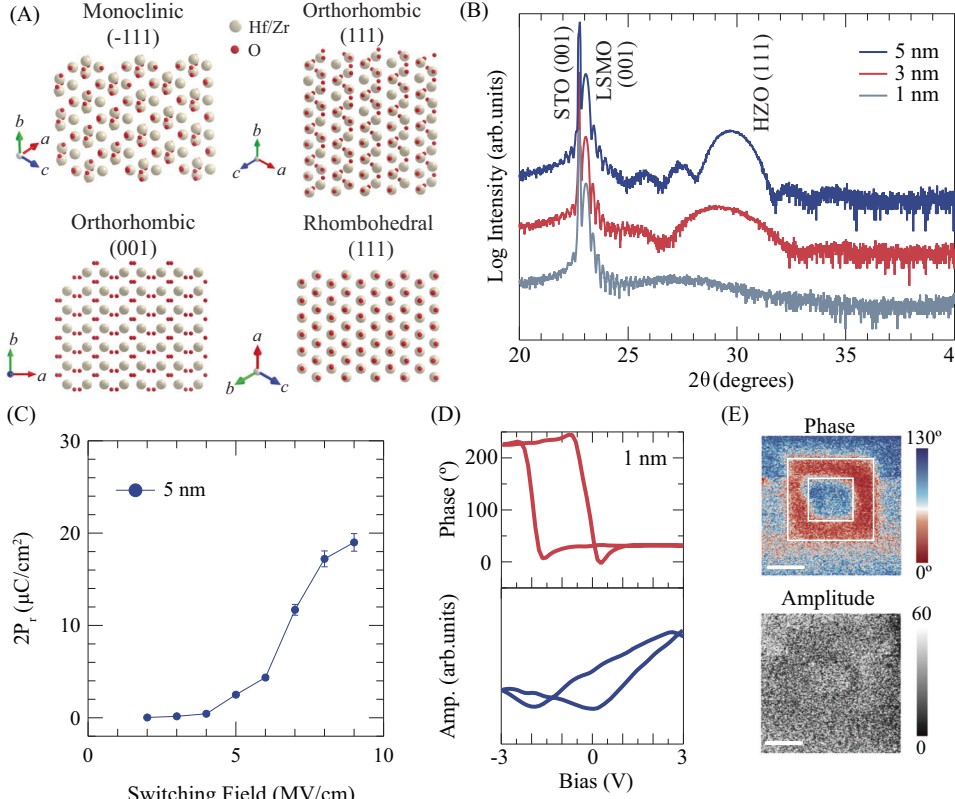

**Fig. 1 | Structural and ferroelectric properties of ultrathin epitaxial HZO films.** **A** HZO polymorph phases with their preferred orientations. The projection direction is perpendicular to the preferred planes. **B** XRD patterns of 1-, 3-, and 5-nm-thick HZO epitaxial films grown on (100) STO substrates buffered with LSMO layers. **C** Switchable polarization of 5-nm-thick HZO epitaxial films evaluated by PUND measurements. The 2Pr values plotted are corrected for the leakage current contribution obtained by analyzing charge profiles of the PUND measurements. Details of the 2Pr value evaluation are provided in Supplementary Materials. The error bars indicate variations in the polarization values obtained by ten repeated measurements. **D** External d.c. bias dependence of off-field PFM phase and amplitude for the 1-nm-thick epitaxial thin films. **E** Box-in-box domain writing for 1-nm-thick HZO epitaxial films with poling bias of ±2 V. The scale bar denotes 500 nm.

in the ultrathin region (<5 nm) where the external factors dominate, can intrinsically maintain metastable phases and ferroelectricity. Thus, fabricating and investigating freestanding membranes of hafnia, which are free from influences of top/bottom electrode materials, are a promising approach for getting an insight into the intrinsic properties of ferroelectric metastable phases of hafnia. Furthermore, stacking membranes of hafnia with the scale-free ferroelectricity on materials that metastable hafnia can not be grown and creating interfaces allow for integrating ferroelectricity into various functional properties like magnetism and exploring potential next-generation devices.

Here, after depositing $Hf_{0.5}Zr_{0.5}O_2$(HZO)/$La_{0.7}Sr_{0.3}MnO_3$(LSMO)/$SrTiO_3$(STO) epitaxial heterostructure by pulsed laser deposition, we fabricated ultrathin (1–5 nm thick) freestanding HZO membranes via selective etching of LSMO layers in hydrochloric acid-based solutions. We found that these ultrathin membranes, which can be transferred onto any material, have ferroelectric polarizations as large as 13 μC/cm² at room temperature, even down to the 1-nm-thick. By employing high-angle annular dark-field (HAADF) imaging in scanning transmission electron microscopy (STEM), we identified the evolution of the metastable phases from the rhombohedral phase to the orthorhombic one in samples with thicknesses increasing from 1 nm to 5 nm. Our results provide experimental evidence of the scale-free ferroelectricity in the rhombohedral hafnia and critical insights into the formation mechanism and structural phase stability of the metastable hafnia.

## Results

### Epitaxial growth of ultrathin ferroelectric HZO thin films

HZO films with thicknesses ranging from 1 nm to 5 nm were deposited on LSMO-buffered (001) STO substrates by pulsed laser deposition (PLD). The thicknesses of the deposited HZO layers were confirmed by X-ray reflectivity measurements (Fig. S1). Previously, it was shown that HZO films with the metastable orthorhombic (or rhombohedral) structure are stabilized through the unique (Hf, Zr)O$_x$ interfacial monolayer produced by Mn-to-Hf/Zr cation exchange during HZO layer deposition[35,36]. Figure 1B shows X-ray 2θ/θ diffraction (XRD) patterns of the fabricated HZO/LSMO/STO heterostructures. Besides the (001) diffraction peaks originating from the LSMO buffers and STO substrates, (111) reflections from the HZO layers appear in the 2θ regions from 27° to 30° and these 2θ values are smaller than that of the commonly seen orthorhombic HZO (o-HZO) in the polycrystalline sample (2θ ˜ 30.5°)[37]. Furthermore, their reflection positions shift toward the lower 2θ side with decreasing thickness, indicating that the interplanar distance along the out-of-plane [111] direction elongates with decreasing thickness. Previous studies have pointed out that such lattice expansion along the [111] direction in HZO films results from increased rhombohedral distortion in HZO[38,39], implying a thickness-dependent evolution of metastable structures of HZO films. No obvious reflections coming from monoclinic HZO films or other secondary impurity phases were observed in our samples, and the half-width full maximum (HWFM) of the (111) reflections for all epitaxial films studied here, which were evaluated by performing omega scans,

are as low as 0.05° (Fig. S1), ensuring the crystallinity of our HZO epitaxial films. Moreover, atomic force microscope (AFM) measurements confirmed that, regardless of the HZO thickness, these ultrathin epitaxial HZO films have smooth surface morphologies with step and terrace structures that are replicas of those of the substrate surface (Fig. S1).

Next, we performed PUND (Positive-Up Negative-Down) measurements to evaluate switchable polarizations in the fabricated ultrathin epitaxial HZO films. The results are displayed in Fig. 1C. For the 5-nm-thick films, the polarization abruptly increases at the electric field at which the polarity of the polarization reverses, and it almost saturates at $2P_r$ ~ 19 $\mu C/cm^2$, at the maximum field we could apply without films' breakdown. We defined the switching field, $E_s$, as the electric field at which the half maximum value of the switched polarization is seen and determined it to be ~6.8 MV/cm for the 5-nm-thick films. The polarization switching behavior matches that expected from ferroelectric hysteresis loops (Fig. S2). On the other hand, the polarizations of the 1-nm- and 3-nm-thick films keep increasing with the electric field due to large leakage currents that possibly stem from oxygen migration between the HZO and LSMO under the external electrical field[40] (Fig. S2), as will be discussed in detail later. To further explore ferroelectricity in the 1-nm-thick HZO films, we performed piezoresponse force microscopy (PFM) characterizations to measure the piezoelectric response characteristic of the ferroelectricity. The out-of-plane piezoelectric response was measured under zero external d.c. bias (off-field piezoelectric response). The results are shown in Fig. 1D, E. The off-field piezoelectric response displays a clear hysteresis loop against sweeping of the external bias with a phase change of around 180°; this is accompanied by butterfly curves in the amplitude-bias plots (Fig. 1D). The horizontal shift of the hysteresis center is probably due to asymmetries between the tip/HZO and HZO/LSMO interfaces[8,41]. Figure 1E shows a ferroelectric domain mapping obtained after box-in-box writing processes. Depending on the polarity of the writing bias, the film displays well-defined ferroelectric domains characterized by distinct phases, which are retained after the domain writing processes (Fig. S3). These results confirm the ferroelectric polarizations in the ultrathin HZO films can be electrically switched.

## Fabrication of ultrathin HZO membranes

Potassium iodide (KI)-diluted hydrochloric acid (HCl) solution was used to etch the LSMO buffer selectively and exfoliate ultrathin HZO films from the substrates, as depicted in Fig. 2A. The exfoliated and freestanding HZO membranes were then transferred to SiO₂/Si substrates by using a conventional dry transfer method[42]. The details of the fabrication process are provided in Fig. S4. As shown in Fig. 2B, the transferred membranes are as large as 5 ×5 mm², which is comparable to the sizes of the STO substrates used for the HZO film growth. These results show the feasibility of our membrane fabrication process. Figure 2C shows the AFM surface morphology of a membrane made from the 5-nm-thick HZO film and transferred onto a SiO₂/Si substrate. The membrane has a smooth and flat surface, with a root mean square (RMS) roughness of 0.38 nm and no obvious residuals of organic stamps. The line profile across the edges of the HZO membranes in Fig. 2C shows that the step height from the substrate surface is around 5 nm, identical to the thickness of the original epitaxial HZO films. The fact that the thickness of the HZO layer is maintained during the exfoliation and transfer processes was confirmed by making AFM measurements on the 1-nm- and 3-nm-thick membrane samples and analyzing the zero loss peak (ZEP) electron energy loss (EEL) spectrum (Fig. S5). We further performed energy-dispersive-x-ray spectroscopy (EDS) characterizations in STEM on these 5-nm-thick HZO membranes on the SiO₂/Si substrates (Fig. S6). Although atomically-resolved cross-sectional HAADF image contrasts in the membrane layers were hard to obtain due to misalignments between the incident electron beams and in-plane directions of the membranes, neither La nor Sr EDS signals were detected from the membrane layers, thereby verifying that the LSMO layers can be completely etched in our exfoliation process. The interface between the transferred membrane layer and the substrate is found to be chemically sharp, and the bottom layer of the membrane consists of Hf and Zr. The weak Mn signal detected in the membrane layer implies that small amounts of Mn, which is exchanged with Hf/Zr during the HZO layer deposition[36], are residual and incorporated into the HZO lattices.

XRD was then used to characterize the structural properties of the HZO exfoliated membranes transferred to the SiO₂/Si substrates. Figure 2D compares 2θ/θ diffraction profiles for the 5-nm-thick membranes and epitaxial films of HZO. Although the diffraction intensities are weak, the membrane samples exhibit diffraction intensities at almost the same position as the (111) reflection of the HZO epitaxial films. While the interfacial structural mismatch and resultant epitaxial strain are considered to help stabilize HZO films with metastable structures, our results indicate that the ultrathin HZO membranes can maintain the metastable phase without external factors like strain, and the observed reflection can be indexed as (111) of HZO. The lattice spacing between the (111) planes of the HZO membrane is still slightly larger than that of the o-HZO in polycrystalline HZO films

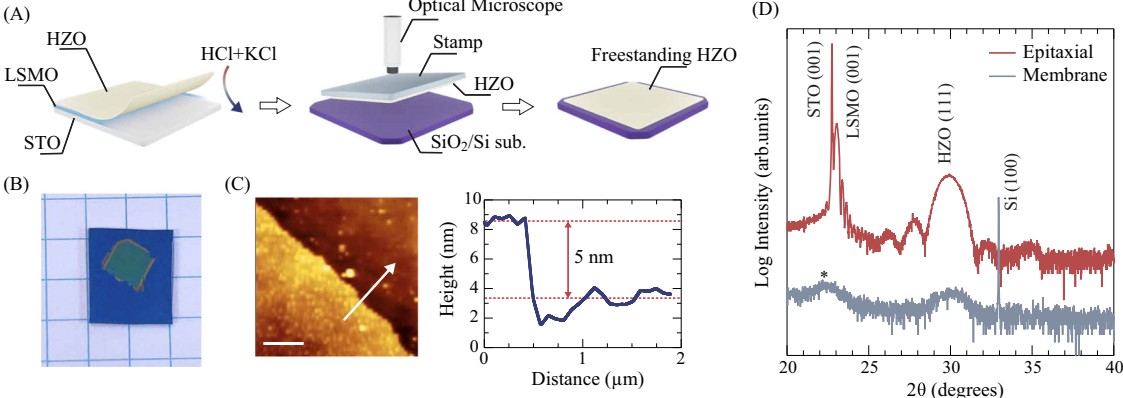

**Fig. 2 | Fabrication and structural properties of freestanding HZO membranes.**
**A** Fabrication process of freestanding HZO membranes. **B** Optical image of photoresist-covered 5-nm-thick HZO membranes transferred onto 1 cm × 1 cm SiO₂-coated (100) Si substrates. **C** AFM surface morphology of the 5-nm-thick HZO membrane and height profile along the white arrow in the surface morphology. The scale bar in the figure denotes 1 μm. **D** XRD patterns of 5-nm-thick HZO membranes transferred on SiO₂/Si substrates. The diffraction pattern from 5-nm-thick epitaxial HZO films is shown for comparison. The broad peak seen around 2θ ~ 22° (marked with the asterisk) comes from the sample holder used in our measurement setup.

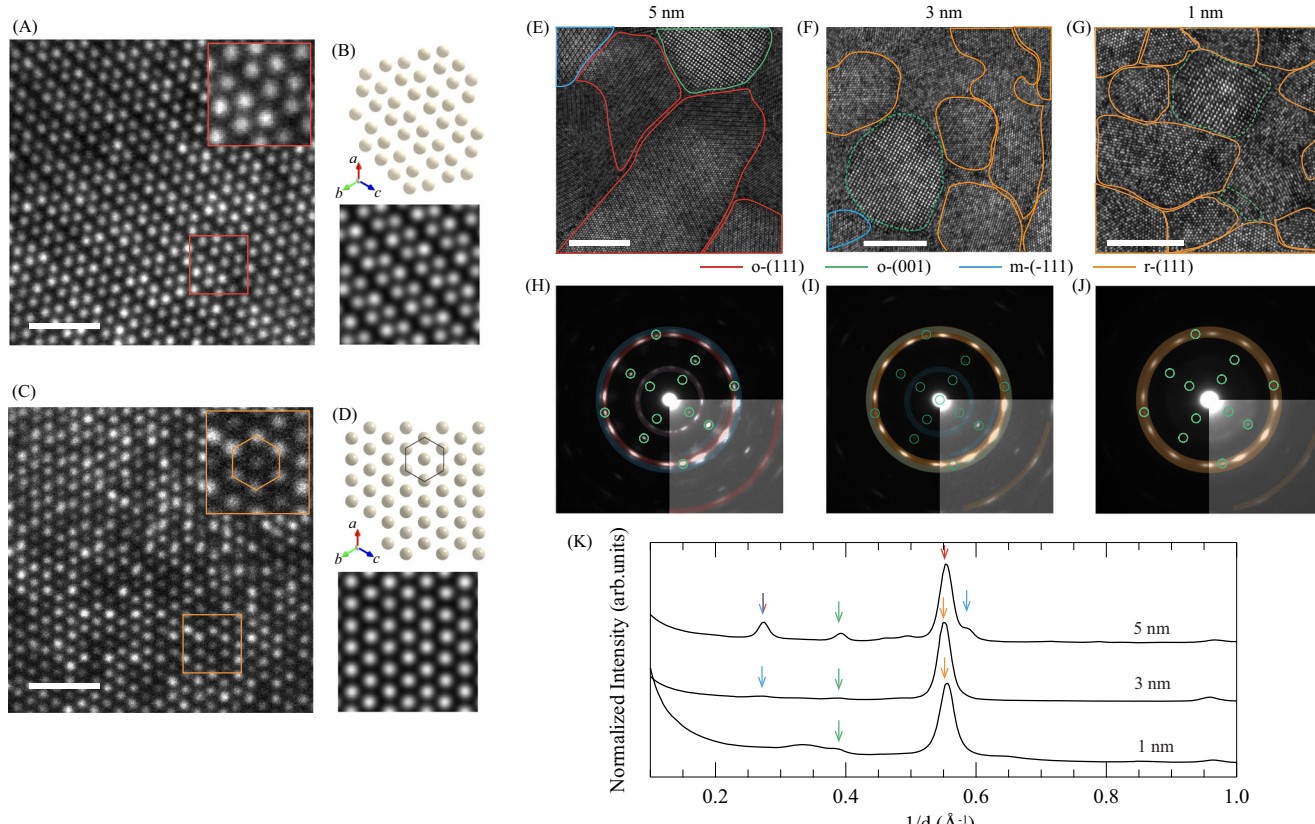

**Fig. 3 | Structural phase identification and distribution analysis of ultrathin HZO membranes. A** Plan-view HAADF-STEM image of a grain with the o-(111) structure in the 5-nm-thick HZO membrane. The inset is the magnified part of the red square frame. **B** Atomic arrangement and simulated HAADF-STEM image of o-(111) HfO$_2$. **C** Plan-view HAADF-STEM image of a grain with the r-(111) structure in the 1-nm-thick HZO membrane. The inset is the magnified part of the orange square frame. **D** Atomic arrangement and simulated HAADF-STEM image of r-(111) HfO$_2$. The scale bars in **A** and **C** denote 1 nm. **E–G** Plan-view HAADF-STEM images and structural phase distribution analysis of (**E**) 5-nm-, (**F**) 3-nm-, and (**G**) 1-nm-thick HZO membranes. The grains having r-(111), o-(111), o-(001), and m-(−111) atomic arrangements are colored in orange, red, green, and blue, respectively. The (001)-oriented grains circled with dashed green lines in **F** and **G** show almost no periodic modulation in the distance between neighboring Hf/Zr atoms characteristic of the o-(001) structure. The grains not circled are ones whose structural phases cannot be identified. The scale bars in **E–G** denote 5 nm. **H–J** SAED patterns for (**H**) 5-nm-, (**I**) 3-nm-, and (**J**) 1-nm-thick HZO membranes. The diffraction patterns are colored with the same color codes as in (**E-G**) to identify which diffraction patterns originate from which structural phases of HZO. Insets are the high-contrasted parts of the SAED. **K** Normalized diffraction intensity profiles versus reciprocal distance extracted from the SAED patterns in **H–J**. The colors of the arrows represent the structural phases that mainly contribute to the arrowed peaks and share the same color code as is used in **E–G**.

($d_{111}$ = 2.89 Å)[37], implying that structural distortions introduced during the HZO epitaxial growth might have remained in the exfoliated membranes. Furthermore, the diffraction profiles around the (111) reflection for the membranes remain unchanged for over two months after being exfoliated (Fig. S7), indicating the robustness of the metastable phase in the exfoliated HZO membranes.

## Crystal structures of ultrathin HZO membranes

To identify the crystal structures and their distribution within ultrathin HZO membranes, we performed plan-view STEM observations and selected area electron diffraction (SAED) measurements on 1, 3, and 5-nm-thick HZO membranes. Main results are summarized in Fig. 3 and details of the structural phase identification are provided in Fig. S8. We found that the membranes consist of grains with several distinct structures, which can be categorized as (−111)-oriented monoclinic (m-(−111)), (111)-oriented orthorhombic (o-(111)), (001)-oriented orthorhombic (o-(001)) and (111)-oriented rhombohedral (r-(111)) structures. We also note that while a tiny amount of amorphous grains (<4%) are detected in the 1-nm-thick membranes (as indicated in Fig. S8C), no amorphous grains are observed for the 3 and 5-nm-thick membranes. These observations ensure the crystallinity of the HZO membranes. Interestingly, the structural phases in the membranes are thickness-dependent. Cation arrangements of the phases identified in plan-view HAADF-STEM images of the membranes are shown in

Figs. 3A–D and S8. The o-(111) phase in 5-nm-thick membranes exhibits periodic modulations in the distance between neighboring Hf/Zr atoms (Fig. 3A), which are in good correspondence with the atomic model of the o-(111) phase and simulated STEM image (Fig. 3B). A structural phase distribution analysis in different regions within the 5-nm-thick HZO membranes (Figs. 3E and S8) shows that a majority (~70%) of the grains (whose cation arrangements can be identified) have the metastable o-(111) phase while a minority have the o-(001) and most stable m-(−111) phases. On the contrary, the cation arrangement of the r-(111) phase in the 1 and 3-nm-thick HZO membrane differs from those of the o-(001), o-(111) and m-(−111) phases, and the distances between neighboring Hf/Zr atoms are almost the same without any of the periodic modulations seen in the 5-nm-thick membranes (Figs. 3C and S8B). Furthermore, the hexagonal cation arrangements in the HAADF-STEM images match the structural model and simulated STEM image of the r-(111) phase (Fig. 3D). Interestingly, the phase distribution analysis for the 3-nm-thick membrane in Fig. 3F shows that more than 80% of the grains have the r-(111) phase structure whose Gibbs energy is even larger than that of the orthorhombic structure[43,44], and the rest of grains have the m-(−111) and o-(001) phases. When the membrane is further thinned down to 1 nm, and more than 90% of the grains have the r-(111) phase structure, and very few m-(−111) and o-(001) grains coexist (Fig. S8C). It should be noted that the minor grains categorized as o-(001) for the 1 and 3-nm-thick

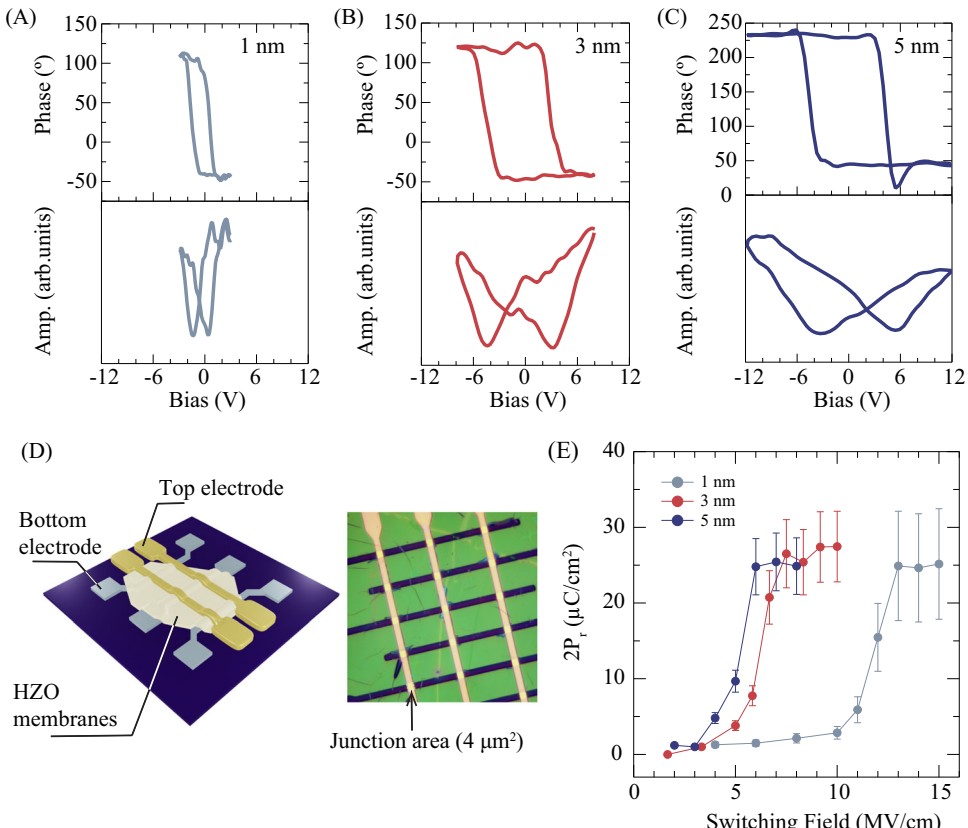

**Fig. 4 | Demonstration of ferroelectricity in ultrathin HZO membranes. A–C** Off-field PFM hysteresis loops of 1-, 3-, and 5-nm-thick HZO membranes. **D** Schematic and optical images of fabricated cross-bar junctions of HZO membranes for PUND measurements. The junction area is 4 μm². **E** Switchable polarization of the 1-, 3-, and 5-nm-thick HZO membranes evaluated by PUND measurements. The 2Pr values plotted are corrected for the leakage current contribution obtained by analyzing charge profiles of the PUND measurements. Details of the 2Pr value evaluation are provided in Supplementary Materials. The error bars indicate variations in the polarization values obtained from 10 repeated measurements.

membranes show almost no periodic modulations in the distance between neighboring Hf/Zr atoms, characteristic of the o-(001) structure, in contrast to those in the 5-nm-thick membrane (Figs. 3E–G and S8). The observed cation arrangements imply that structure candidates of the (001)-oriented grains in the 1 and 3-nm-thick membranes are cubic, tetragonal, or rhombohedral. As shown in Fig. S8b, cation arrangements and SAED patterns of these structures projected in the [001] direction are essentially the same and identifying the structural phase of the (001)-oriented grains in the 1 and 3-nm-thick membranes requires further characterizations.

The SAED patterns in Figs. 3H–J and S9 confirm that the HZO membranes have a thickness-dependent structural phase distribution. The SAED patterns for the 5-nm-thick membranes (Fig. 3H), including weak reflection spots, can be regarded as a combination of diffraction patterns from the o-(111) phase (red), o-(001) phase (green) and m-(−111) phase (blue) based on the simulation (Fig. S9), which is consistent with the results of the phase identification and phase distribution analysis based on the HAADF-STEM images. The SAED patterns of 1 and 3-nm-thick HZO membranes (Fig. 3I and 3J) and their analysis based on the SAED simulation (Fig. S9) support that the majority of the grains in both membranes have the r-(111) structure. Figures 3K and S10 plot the SAED intensity against the reciprocal distance determined from the diffraction patterns in Fig. 3H–J, further validating the results of the phase distribution analysis. The structural models with the bulk (or computed) lattice parameters reproduce the positions of most of the reflections. Our results thus indicate that the metastable phases in HZO can be sustained without external factors such as epitaxial strain.

## Ferroelectricity in Ultrathin HZO Membranes

We used PFM to explore the ferroelectricity in ultrathin HZO membranes. In order to apply a d.c. bias along the out-of-plane direction of the membranes, the membranes were transferred onto Pt/Au-sputtered SiO₂/Si substrates. Figures 4A–C plot typical out-of-plane off-field piezoelectric responses for 1-, 3-, and 5-nm-thick HZO membranes against the external d.c. bias at room temperature. All the membranes display butterfly-shaped amplitude-hysteresis loops with 180° phase changes upon polarization switching, which are the piezoelectric responses characteristic of conventional ferroelectrics. These observations demonstrate that the ultrathin HZO membranes can maintain ferroelectricity at room temperature even down to the 1-nm-thick. While a finite imprinting field is seen in all the HZO membranes, its magnitude remains almost unchanged against changes in both the measurement cycles and bias sweep directions (Fig. S11), implying that the effect of charge injection on the piezoelectric response is negligibly small. In addition, the d.c. bias voltage necessary for switching the ferroelectric polarization increases with the membrane thickness. We also made ferroelectric domain mappings of the ultrathin HZO membranes, whose ferroelectric domains with upward and downward polarizations can be written by poling the membranes (Fig. S12). Our PFM results show intrinsic scale-free ferroelectricity in the HZO membranes.

To corroborate our finding of ferroelectricity in the ultrathin membranes, we employed PUND measurements. To minimize the effect of leakage component, we fabricated cross-bar junctions whose measurement area is 4 μm² by transferring HZO membranes onto Pt bottom electrodes that had been patterned into stripes and then

thermally depositing Au stripe electrodes on top of the membranes, as depicted in Fig. 4D. An optical image of the fabricated cross-bar junctions is presented in Fig. 4D. Figure 4E shows the field dependence of the switchable polarizations for the membranes. Details of our PUND measurements and their analysis, including non-ferroelectric (leakage current) contribution estimation from the measured charge profile, are provided in Fig. S13. The polarizations of all the membranes abruptly increase and become saturated at $2P_r \sim 25 \pm 5$ $\mu C/cm^2$ with increasing electric field, indicating that $P_r$ is independent of the membranes' thickness. It should be noted that the extracted switchable polarization also stays steady against the looping of electrical fields (Fig. S13), indicating that the leakage component is less important in the extracted switchable polarization. Moreover, the switchable polarizations with large switching field can also be reproduced on other junctions fabricated using 1-nm-thick hafnia membranes (Fig. S14), further validating the survival of ferroelectricity in 1-nm-thick hafnia. An increase in $E_c$ with decreasing HZO thickness, which is in agreement with the results of the PFM characterizations, has also been seen in experiments on ferroelectric tunnel junctions with barrier layers of ultrathin HZO[29,45,46]. The observation of the robust switchable polarization in the 1-nm-thick rhombohedral HZO membranes indicates the scale-free nature of the ferroelectricity in hafnia, regardless of its metastable phase (rhombohedral or orthorhombic)[47]. Moreover, $E_s$ of the membranes is lower than those of epitaxial films: it is 5.1 MV/cm for the 5-nm-thick membrane and 6.8 MV/cm for the epitaxial thin film, respectively. The lowering in Es for the membrane implies that substrate-induced effects, such as lattice clamping and strain, affect domain wall motions associated with the polarization reversal, as seen in perovskite ferroelectrics[16,48]. More importantly, given that the polarization of HZO originates from the oxygen displacements and would be influenced by oxygen diffusion across the HZO/electrode interfaces under electric fields[40], the observed differences in the ferroelectric properties between the freestanding membranes and epitaxial films of HZO possibly stem from the oxygen injections and extractions from bottom LSMO layers, which are known to be served as an oxygen sink against the HZO layer. Thus, utilizing oxygen-inactive metal electrodes and suppressing oxygen diffusion across the HZO/electrode interface are critical for maintaining robust ferroelectricity in the ultrathin HZO membranes.

## Stability of ferroelectric phase in hafnia

Our results demonstrate that the metastable and ferroelectric phases of hafnia can be stable without the help from any external factors, such as strain and interfacial effects that are believed to be crucial for forming metastable phases of hafnia[38,49–51], implying that energy barriers between hafnia's distinct phases are rather high[52]. Therefore, hafnia with metastable structures and scale-free ferroelectricity can be fabricated into the membrane form free of strain and interfacial effects and transferred onto any material.

Another thing we should point out is that the structural phase in the membranes evolves from rhombohedral to orthorhombic with increasing thickness. Given that the rhombohedral phase had not been identified in hafnia's phase diagram until it was found to be stabilized through epitaxial growth[38] and its Gibbs energy is higher than the metastable orthorhombic phase[43,44], the rhombohedral-to-orthorhombic structural transformation can be understood to be a result of structural energy relaxation of the rhombohedral phase. This scenario can also explain why the rhombohedral structure appears as an almost single phase in the 1-nm-thick membranes, while the orthorhombic phase appears in a mixture with the most stable monoclinic phase in the 5-nm-thick membranes. The smaller grain size of the rhombohedral hafnia than that of the orthorhombic phase is another indication that the rhombohedral phase is less stable than the orthorhombic one, leading to the occurrence of a structural

transformation due to the structural energy relaxation as the thickness (or grain size) increases. Although atomic arrangements between HZO and the structurally dissimilar perovskite LSMO layers have not been fully identified[53], our results indicate that HZO films deposited on LSMO layer is initially grown into the rhombohedral phase through interfacial structural matching and then evolve into the orthorhombic phase, providing hints on understanding of the formation mechanism and phase stability of hafnia.

In summary, by fabricating and characterizing ultrathin hafnia membranes, we reveal the key role of the rhombohedral phase in the scale-free ferroelectricity in hafnia with an out-of-plane polarization of ~13 $\mu C/cm^2$ at room temperature. We further show that the rhombohedral phase relaxes to another metastable orthorhombic phase that appears in a mixture with the most stable monoclinic one. Our results offer critical insights into the ferroelectricity in hafnia and reveal the potential of using ultrathin hafnia membranes to make heterostructures consisting of structurally dissimilar materials that cannot be obtained in conventional film-growth techniques.

## Methods

### Epitaxial thin film growth and membrane fabrication

All epitaxial thin films were fabricated by using pulsed laser deposition (PLD) on $TiO_2$-terminated (100) $SrTiO_3$ (STO) substrates (Shinkosya Co., Japan) with the Coherent COMPex Pro 205 KrF excimer laser ($\lambda = 248$ nm). $La_{0.67}Sr_{0.33}MnO_3$ (LSMO) sacrificial layers were deposited at 650 °C under an oxygen partial pressure of 100 mTorr to obtain optimal crystallinity. During the deposition, the LSMO ceramic targets were ablated with a laser density of 1.2 $J/cm^2$ and at the repetition frequency of 5 Hz. $Hf_{0.5}Zr_{0.5}O_2$ (HZO) thin films were subsequently deposited on the sacrificial layers without breaking the vacuum of the PLD chamber. The HZO was deposited at 800 °C and under an oxygen partial pressure of 75 mTorr. The HZO targets were pulsed with a laser density of 1.4 $J/cm^2$ at 2 Hz. After the depositions, the samples were cooled to room temperature under an oxygen partial pressure of 75 mTorr.

HZO membranes were obtained by etching the LSMO layers in water solutions containing 5 wt.%. hydrochloric acid (HCl) and 0.05 M potassium chloride (KCl). After the exfoliation processes, the membranes were transferred onto adaptive substrates ($SiO_2$-coated Si substrates and Au-coated $SiO_2$/Si substrates) with the conventional dry transfer method described below (Fig. S4). Organic protection layers (photoresist LOL2000, Microposit) were first spin-coated on the surface of epitaxial HZO thin films and then baked at 130 °C for 5 min to increase the adhesion between the HZO and the photoresist layer. The spin-coating was conducted at a rotation speed of 4500 rpm for 1 min, producing 150–200-nm-thick protection layers. After this photoresist coating step, the HZO/LSMO/STO stacks were immersed in the HCl-KI solution. The HCl concentration was optimized to fully etch LSMO for around 36 h and minimize damage to the HZO thin films during the etching process. Once the LSMO layers were etched, the photoresist-coated HZO membranes sat on STO substrates (instead of floating up in the HCl-KI solutions). The membrane samples were then rinsed with deionized water several times. In the final rinsing, HZO membranes that sat on the STO substrates were left in the deionized water for at least 1 h to dissolve the residual KCl fully. The membranes were then gently moved out of the deionized water and dried for 15 min in the ambient atmosphere. After the membrane samples had fully dried, polydimethylsiloxane (PDMS; Gel Pak PF series X17) sheets were attached to the top of the photoresist protection layer at room temperature and mounted up swiftly, leaving the photoresist/HZO stacks sticking to the PDMS sheets. The whole PDMS/photoresist/HZO stacks were then placed on $SiO_2$/Si substrates at 85 °C for 15 min to enhance the adhesion between the HZO membranes and the adaptive substrates. The PDMS was then slowly peeled off at 65 °C, leaving

photoresist/HZO membranes on the adaptive substrate. The photoresist was then dissolved in a developer solution (CD-26, Microposit). The HZO membrane surfaces were cleaned in oxygen plasma treatments, whose conditions were optimized so not to degrade HZO's ferroelectricity. The oxygen plasma cleaning ensured that the transferred membranes had a clean surface, as shown in the AFM topography (Fig. S4B). Optical images of HZO membranes transferred onto $SiO_2$/Si substrates (Fig. S4C) showed clear and different optical contrasts between the membranes and substrates. All fabricated membranes had crack-free surfaces, regardless of thickness. These results validate our exfoliation and transfer processes.

The surface topography of the epitaxial films and membranes of HZO were evaluated with a tapping-mode atomic force microscope (Hitachi AFM5200II).

### Structural characterization

X-ray $2\theta/\theta$ diffraction measurements were performed with a lab-source four-circle diffractometer (X'Pert MRD, PANalytical) using the Cu $K_{\alpha1}$ radiation.

Cross-sectional TEM samples were prepared by FIB-SEM (JEOL JIB-4700F). STEM-EDS elemental mappings were conducted at room temperature on a spherical aberration-corrected STEM (JEOL ARM-200F) equipped with an EDS spectrometer (JED-2300T). The experiments were performed at 200 kV.

For the plan-view STEM observations, ultrathin HZO membranes on a holey carbon film supported by a Cu grid were prepared. (S)TEM measurements were performed at 200 kV on a JEOL 2400FCS equipped with a probe aberration corrector. For the HAADF-STEM imaging, the probe forming semi-angle was set to be 22 mrad and the HAADF detector collection angle was 68-280 mrad. To minimize image distortion due to sample-stage drift and improve the signal-to-noise ratio, 30 HAADF images were sequentially acquired with a fast dwell time of approximately 2 μs/pixel with the same field-of-view. They were then spatially stacked and averaged. HAADF-STEM image simulations were performed using abTEM code[54] by assuming the same accelerating voltage and probe forming semi-angle as in the experiments and corresponding sample thicknesses (i.e., 5 nm for the orthorhombic phase and 1 nm for the rhombohedral phase). To account for the finite size of an electron source, the calculated STEM images were convolved with a two-dimensional Gaussian with a FWHM of 0.08 nm. The SAED patterns and their intensity profiles against reciprocal distances were calibrated using a single crystal of $SrTiO_3$ with lattice parameters precisely determined by X-ray diffraction and simulated with the Single Crystal and CrystalDiffract software (CrystalMaker Software Limited). The simulation parameters were identical to those used for the experimental imaging and SAED acquisition mentioned above. The crystal profiles used for the STEM/SAED simulations are shown in Table S1.

### Ferroelectric property and piezoelectric response characterization

Polarization hysteresis loops and PUND (positive-up-negative-down) measurements were carried out with ferroelectric testers (FCE-1 and FCE10, TOYO Corp.). To evaluate the polarization of the HZO epitaxial films on the LSMO layers, which can be used as bottom electrodes, Au top electrodes with a diameter of 50 μm were thermally deposited and patterned with the conventional lift-off process. Cross-bar junctions were used to characterize the polarization of the HZO membranes. To fabricate cross-bar devices with junction areas of 4 μm², 2-nm-thick Pt bottom electrodes were sputtered on the $SiO_2$-coated Si substrates in a 2 μm-wide stripe pattern. HZO membranes were then transferred to the striped bottom electrodes. Finally, 2-μm-wide Au electrode stripes were thermally evaporated and patterned photolithographically on top of the membranes.

The piezoelectric response and ferroelectric domain structure of the epitaxial films and membranes of HZO were observed by piezoresponse force microscopy (PFM) (Asylum Research, MFP-3D). The responses were collected in dual a.c. resonance tracking (DART) mode with a driving voltage of 0.5 V. For measuring the d.c. bias dependence of the piezoelectric response, the signals were collected when the bias was turned off (the off-field piezoelectric response) after the poling process. Examples of voltage sequences used for measuring the d.c. bias dependence of the off-field piezoelectric response are provided in the supplementary information (Fig. S11).

### Reporting summary

Further information on research design is available in the Nature Portfolio Reporting Summary linked to this article.

## Data availability

The data that support the plots within this article and other findings of this study are available from the corresponding author upon reasonable request.

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

## Acknowledgements

This work was partly supported by Grants-in-Aid for Scientific Research (No. 19H05816 (D.K.), 19H05823 (YU.S.), 21H01810 (D.K.), 21K18196 (S.K.), 22J01665 (K.O.), 22H01523 (T.Y.), 23KJ1239 (YF.S), and 23H05457 (YU.S.)) and by grants for the Integrated Research Consortium on Chemical Sciences and the International Collaborative Research Program of the Institute for Chemical Research in Kyoto University from the Ministry of Education, Culture, Sports, Science, and Technology (MEXT) of Japan. The work was also supported by the Japan Science and Technology Agency (JST) as part of the Advanced International Collaborative Research Program (AdCORP), Grant No. JPMJKB 2304 (YU.S.) and Adopting Sustainable Partnerships for Innovative Research Ecosystem (ASPIRE), Grant No. JPMJAP2312 (D.K., T.Y.) and JPMJAP2314 (YU.S.). The work was also supported by JSPS under the Joint Research Program implemented in association with SNSF (Grant No. JPJSJRP20221504; T.Y.), and MEXT Program: Data Creation and Utilization Type Material Research and Development Project (Grant No. JP- MXP1122683430; T.Y.).

## Author contributions

YF.S. and D.K. conceived the idea and the project plan. YF.S. fabricated samples and performed structural and ferroelectric property characterizations with help from D.K., X.Y., and T.Y.; M.H. carried out cross-sectional STEM observations and EDS elementary mappings. K.O. and S.K. performed plan-view STEM observations. D.K. and YU.S. supervised the project. All authors discussed the experimental data and YF.S. and D.K. co-wrote the manuscript with feedback from all the authors.

## Competing interests

The authors declare no competing interests.
