## [Peer Review File · Nature Communications]

Ferroelectric freestanding hafnia membranes with metastable rhombohedral structure down to 1-nm-thickREVIEWER COMMENTS

Reviewer #1 (Remarks to the Author):

Y. Shen et al. report on the deposition and properties of ultrathin $\text{Hf}_{0.5}\text{Zr}_{0.5}\text{O}_2$ (HZO) membranes. The preparation process involves the epitaxial growth of HZO films with thickness values between 1 and 5 nm on LSMO-buffered $\text{SrTiO}_3(001)$, followed by the chemical etching of LSMO and the transfer of the resulting membranes onto Si/SiO₂ substrates. The films presented contain large fractions of the rhombohedral or orthorhombic polymorphs, and show signatures of out-of-plane ferroelectricity both before and after being transferred.

In my opinion, the paper combines two topics of the outermost interest: ferroelectricity in hafnia-based films, and free-standing membranes of functional oxides. As far as I know, ultrathin hafnia-based membranes as the ones studied here have not been reported before, so the results are novel and will attract a wide audience. The experiments described are careful and thorough, and most conclusions are well supported. The TEM analysis has a high quality.

However, I have a few observations that the authors could address to improve the manuscript:

- (1) The basic interest or the potential applications of the HZO membranes studied here are not sufficiently explained. I suggest a few additional sentences in the Introduction clarifying this topic.
- (2) The authors claim that the HZO films have 1, 3 and 5 nm of thickness, but do not show any proof of that. This can be especially relevant in the thinnest films. (Fig. S6 is valuable, but insufficient.) I suggest the authors to show X-ray reflectivity measurements (and fits) and/or TEM cross-sections of either the $\text{SrTiO}_3/\text{LSMO}/\text{HZO}$ samples or the $\text{Si}/\text{SiO}_2/\text{HZO}$ stacks for the three values of thickness studied.
- (3) Related to the previous point, the word “monolayer” appears several times throughout the paper, including the title and abstract. This term is not accurate, as the lowest thickness explored (1 nm) corresponds to around 2 unit cells of HZO.
- (4) The result that the 1 nm-thick films contain almost exclusively [111]-oriented rhombohedral phase, and the 5-nm ones show a large mixture of phases and orientations is remarkably interesting. Could the authors provide some more structural details on the 3 nm-thick films?
- (5) The XRD results presented in Fig. 2(D) and Fig. S7 show a broad band at $22\text{-}23^\circ$ in the 5 nm-thick HZO membrane which cannot be attributed to any hafnia polymorph. The authors should explain its origin. Does it correspond to STO or LSMO, suggesting that the membrane still contains rests of a perovskite oxide and thus is not properly free-standing? Are the results similar for other thickness values?
- (6) The identification of the structures of different grains is based on the comparison between the STEM-HAADF experimental images and simulations. Showing other polymorphs and zone axes giving similar images would be useful to reinforce the structural assignment done in the text.

(7) Strictly speaking, piezoresponse proves the piezoelectric behavior of the films, not necessarily their ferroelectric nature. "Piezoelectricity" could be used instead of "Ferroelectricity" most times in the section beginning in line 191, including its title, and also in Section 6 of the Supplementary Materials.

(8) The authors should explain why measuring polarization-field hysteresis curves was not possible in some of the films. For instance, loops similar to Fig. S2 could be shown in Fig. 1 for the 1 nm and 3 nm-thick samples.

Minor suggestions, typos, etc.:

(a) I would prefer the term "crystal structure" instead of "atomic structure" in lines 154, 156 and following.

(b) The crystal directions and orientations should be indicated as [hkl] instead of (hkl). The XRD reflections should be indicated as hkl instead of (hkl).

(c) "Poling" should be used instead of "polling".

(d) The paper with DOI 10.1021/cm021111v is an important one in the field of epitaxial stabilization of oxide films. It can be added together with Refs. 9-11.

(e) The length of the scale bars should be clarified in all the figures, at least Figs. 2C, S3, S6A and S11A.

(f) (E) label is missing in the caption of Fig. 4.

(g) Fig. S9(C) is mentioned instead of Fig. S12(C) in the last paragraph of section 7 of the Supplementary Materials.

(h) Ref. 2 in the caption of Table S1 is likely Ref. 1.

(i) "Topography" should be used instead of "Topology" in Fig. S12(A).

Reviewer #2 (Remarks to the Author):

The authors reported ferroelectric freestanding hafnia membranes with metastable rhombohedral structure down to 1 nm. The result is interesting. However, the reviewer believes that accurately evaluating electrical and physical properties of 1 nm-thick films must be more accurately evaluated with the state-of-the-art techniques. The reviewer can agree that it is highly challenging, but it is need for the publication of the work in highly esteemed journal such as Nature Communications. Furthermore, some discussions were not clear for the reviewer considering the recent understanding of ferroelectric HZO films. Therefore, the reviewer believed that this work can be reconsidered after a mandatory major revision. The detailed comments are attached below.

In the title, what is the meaning of monolayer? 1 nm is already 2 unit cells thick with containing four metal layers and four oxygen layers. Clarify this point.

What is the rhombohedral space group considered in this work? There have been reports on epitaxial HZO films epitaxially grown using PLD. However, there is a recent report on metal-rich rhombohedral phase published in Science. Please clarify this point.

The authors wrote that “Furthermore, their reflection positions in 2θ shift toward the lower 2θ side with decreasing thickness, revealing large lattice expansions along the out-of-plane [111] direction in the monolayer limit.” However, what is the evidence for this statement? The reviewer believes that a further analysis evaluating both out-of-plane and in-plane lattice parameters is needed.

The switching field in figure 1c seems extremely high especially for 1 and 3 nm-thick film. It is reaching the field required for the electron tunneling already. Can we believe that the measured switching polarization is free from the charge injection and tunneling? The reviewer has the same concern on figure S11.

It was not clear for the reviewer whether the HZO films were completely crystallized from figure 2b. It is well known that the crystallization of HZO films thinner than 3 nm. Please provide more analysis or explanation on the crystallization degree of the HZO films.

The spatial phase map and selective area electron diffraction in figures S8 and S9 is very interesting. However, what is the accuracy of this analysis? Provide more details of the analysis. Which phases are the most difficult phase to distinguish from each other? How accurately does the phase analysis differentiate the relative position of oxygen ions which should be the key to differentiate the different crystallographic structure of HZO films? It was already very tough to evaluate the 10 nm-thick films using these techniques. How did the authors improve this analysis applicable to 1 nm membrane? Please compare the analysis in this work to that in previous works.

The result in figure S10 was difficult to understand. Why is there no consideration of orthorhombic Pca21 in red color for the result of 1 nm-thick film in the bottom panel?

Reviewer #3 (Remarks to the Author):

This work reveals the ferroelectric properties of HZO membranes down to atomic thickness. It also points out the transformation from rhombohedral phase to orthorhombic phase when the film thickness increase. Overall the result is very useful for the development of 2D ferroelectrics. My main concern is regarding the ferroelectric properties extracted from PUND tests.

1) There is only one P-E loop demonstrated, in Figure S2 of the SI, but no such result shown in the main text. Even though this figure is related to the thick 5 nm case, the loop is not well saturated. Given that the authors claim ferroelectricity from 1 nm to 5 nm, then P-E loop for each thickness should be provided.

2) The $2Pr$ values extracted from PUND tests are questionable. From Figure S2 it is shown

that the loop is not saturated, with giant possible leakage current. It is well-known that leakage current can be large in ultra-thin films. On the other hand, the PUND result should eliminate the influence from these leakage. However, the reported $2Pr$ value from PUND in the main text exactly correspond to the $2Pr$ value extracted from the leakage-containing P-E loops, and without considering the retention loss (there is a considerable gap in the P-E loop, indicating retention problem). Hence, the argument for such high $2Pr$ values from PUND is questionable. A reliable $2Pr$ estimation must be given based correctly on PUND results.

Comments from Reviewer #1 (Remarks to the Author):

(#1-0) Y. Shen et al. report on the deposition and properties of ultrathin $\text{Hf}_{0.5}\text{Zr}_{0.5}\text{O}_2$ (HZO) membranes. The preparation process involves the epitaxial growth of HZO films with thickness values between 1 and 5 nm on LSMO-buffered $\text{SrTiO}_3(001)$, followed by the chemical etching of LSMO and the transfer of the resulting membranes onto Si/SiO₂ substrates. The films presented contain large fractions of the rhombohedral or orthorhombic polymorphs, and show signatures of out-of-plane ferroelectricity both before and after being transferred.

In my opinion, the paper combines two topics of the outermost interest: ferroelectricity in hafnia-based films, and free-standing membranes of functional oxides. As far as I know, ultrathin hafnia-based membranes as the ones studied here have not been reported before, so the results are novel and will attract a wide audience. The experiments described are careful and thorough, and most conclusions are well supported. The TEM analysis has a high quality.

Author's response:

We appreciate the review's agreement to the novelty of our work. We also thank reviewer#1 for suggestions to improve our work. All your comments have been addressed as below, and the manuscript has been revised. The revised part of the manuscript is colored in red. We believe that our work is now greatly improved and more complete. Thus, we hope our work can be recommended for publication in *Nature Communications*.

(#1-1) The basic interest or the potential applications of the HZO membranes studied here are not sufficiently explained. I suggest a few additional sentences in the Introduction clarifying this topic.

Author's response:

In response to the reviewer's comment, we have added in the revised version the sentences describing the basic interest and potential application of HZO membranes (Main, Page 2 Line 27-33)

Changes from the original manuscript:

i). (Main, Page 2 Line 27-33) ..., can intrinsically maintain metastable phases and ferroelectricity. Thus, fabricating and investigating freestanding membranes of hafnia, which are free from influences of top/bottom electrode materials, are a promising approach for getting an

insight into the intrinsic properties of ferroelectric metastable phases of hafnia. Furthermore, stacking membranes of hafnia, which has scale-free ferroelectricity, on materials that metastable hafnia can not be grown on and creating interfaces with using conventional thin-film techniques allows for integrating ferroelectricity into various functional properties like magnetism and exploring potential next-generation devices.

(#1-2) The authors claim that the HZO films have 1, 3 and 5 nm of thickness, but do not show any proof of that. This can be especially relevant in the thinnest films. (Fig. S6 is valuable, but insufficient.) I suggest the authors to show X-ray reflectivity measurements (and fits) and/or TEM cross-sections of either the SrTiO₃/LSMO/HZO samples or the Si/SiO₂/HZO stacks for the three values of thickness studied.

Author's response:

We performed X-ray reflectivity (XRR) measurements and their fitting analysis for 1, 3, and 5nm-thick HZO epitaxial films grown on LSMO-buffered SrTiO₃ substrates. The results, shown in Fig. R1 below (labeled as Fig. S1 (A) in Supplementary Materials), confirm that our fabricated HZO layer has the same thickness as the designed value.

We would also like to emphasize that our AFM measurements confirmed that the thickness of our HZO membranes is basically the same as the one for the original HZO epitaxial film. The results of our AFM-based thickness evaluation were provided in Section 3 and Fig. S5 in Supplementary Materials. To cross-check the thickness of the HZO membrane, we further evaluated the thickness of our 1-nm-thick membranes with an electron energy loss (EEL) spectrometer. The results have been provided as Fig. R2 (labeled as Fig. S5 (D-F) in Supplementary Materials), the ZLP EELS analysis shows that the thickness of our thinnest transferred membranes should be thinner than 1.55 nm, ensuring that the thinnest HZO membrane is around 1 nm thick and that the HZO layers maintain their thicknesses during the exfoliation and transfer processes. The detailed analysis is provided in Supplementary Materials, and the manuscript has been revised accordingly.

Fig. R1. XRR profiles and their simulation results for epitaxial thin films with different thickness. **(labeled as Fig. S1 (A) in Supplementary Materials)**

Fig. R2. Thickness analysis from electron energy loss spectroscopy. A) ZLP EEL spectrum obtained from a 1 nm HZO membrane. (B) ZLP EEL spectrum from (A) on the semi- logarithmic scale. (C) EEL spectrum near the C K-edge of the 1 nm HZO membrane. **(labeled as Fig. S5 (C-E) in Supplementary Materials)**

Changes from the original manuscript:

i). Add Fig. R1 to the manuscript (Supplementary Materials, labeled as Fig. S1(A)) with the description: **(Supplementary Materials Section 1; Page 2 Line 3-7)** The thickness of the HZO epitaxial layers grown on (La,Sr)MnO₃-buffered SrTiO₃ substrates (before exfoliation processes) was evaluated by X-ray Reflectivity (XRR) measurements. The XRR profiles for the HZO layers whose thickness was set to be 1, 3, and 5 nm are shown in Fig. S5(A). The observed features in the profiles are well reproduced by assuming the HZO thickness as designed.

ii). Add Fig. R2 to the manuscript (Supplementary Materials, labeled as Fig. S5(D-F)) with description: **(Supplementary Materials Section 3; Page 3 Line 13-21)** Furthermore, to cross-check the thickness of the membranes, we evaluated the thickness of the thinnest (1-nm-thick) membrane by using the electron energy loss (EEL) spectrum. Fig. S5(C) shows the zero loss peak (ZLP) EEL spectrum obtained from the HZO membrane. To clearly see the plasmon intensity, the ZLP data in Fig. S5(C) are re-plotted on the semi-logarithmic scale in Fig. S5(D). The relative thickness, t/λ , was calculated to be 0.0189. The λ value for Hf_{0.5}Zr_{0.5}O₂ is 82.28 nm, giving the thickness t of 1.55 nm. As shown in Fig. S5(E), a slight signal at the C K-edge indicates the presence of minor contamination in the membrane sample, likely due to electron beam irradiation and sample handling. Therefore, the obtained thickness (1.55 nm) would be overestimated and slightly larger than the actual thickness.

(#1-3) Related to the previous point, the word “monolayer” appears several times throughout the paper, including the title and abstract. This term is not accurate, as the lowest thickness explored (1 nm) corresponds to around 2 unit cells of HZO.

Author’s response:

As pointed out by the reviewer, we admit that the term “monolayer” is confusing. We have replaced the “monolayer” with “ultrathin” or “1-nm-thick”.

(#1-4) The result that the 1 nm-thick films contain almost exclusively [111]-oriented rhombohedral phase, and the 5-nm ones show a large mixture of phases and orientations is remarkably interesting. Could the authors provide some more structural details con the 3 nm-thick films?

Author’s Response:

In response to the reviewer's comment, we performed plane-view STEM observations and structural phase distribution analysis for the 3-nm-thick HZO membrane. The phase distribution, SAED patterns, and their diffraction intensity profiles have been provided in Fig. R3-R4 (Figs. 3 (F), 3(I) and 3(K) in the revised version). It can be seen that as expected from the STEM results from the 1 and 5 nm thick membranes, the 3-nm-thick HZO membrane consists of a mixture of a majority phase of r-(111) (colored in orange) and minority phases of m-(-111) (blue) and o-(001) (green). The results strengthen our conclusion that the rhombohedral to orthorhombic structural transformation originates from a structural energy relaxation of the rhombohedral phase associated with the increase in thickness. We also found from careful structural analysis that the minor grains categorized as o-(001) in the 1 and 3-nm-thick membranes show almost no periodic modulation in the distance between neighboring Hf/Zr atoms characteristic of the o-(001) structure, in contrast to those in the 5-nm-thick membrane (Figs. 3E-G and S8). The observed cation arrangements imply that structure candidates of the (001)-oriented grains in the 1 and 3-nm-thick membranes are cubic, tetragonal, and rhombohedral. However, as shown in Fig. S8b, cation arrangements of these structures projected in the [001] direction are essentially the same, and identifying the structural phase of the (001)-oriented grains in the 1 and 3-nm-thick membranes requires further characterizations. In the revised manuscript, we keep sorting this type of the minority grains as o-(001) for simplicity, and this sorting does not change our main conclusion. Details of STEM observations and analysis for the 3-nm-thick membrane have been included in the Supplementary Materials. The manuscript has also been revised accordingly.

Fig. R3. Phase distribution and SAED patterns of the 3-nm-thick HZO membrane. The scale bar denotes 1 nm. The rings indicate r-(111), o-(001), and m-(-111) are colored in orange, green, and blue, respectively. The (001)-oriented grains circled with dashed green lines show almost no periodic modulation in the neighboring Hf/Zr distance characteristic of the o-(001) structure. **(Labeled as Figs. 3 (F) and 3(I) in the manuscript)**

Fig. R4. Normalized diffraction intensity profiles versus reciprocal distance extracted from the SAED patterns in Fig. R3. The diffraction peaks indicated with the orange, green, and blue arrows originate from the r-(111), o-(001), and m-(-111), respectively. **(Included in Fig. 3(K))**

Changes from the original manuscript:

i). Incorporated Figs. R3 and R4 into Fig. 3, with additional descriptions: **(Main, Page 5, Line 22-34)** Interestingly, the phase distribution analysis for the 3-nm-thick membrane in Fig. 3(F) shows that more than 80% of the grains have the r-(111) phase structure whose Gibbs energy is even larger than that of the orthorhombic structure^{43,44}, and the rest of grains have the m-(-111) and o-(001) phases. When the membrane is further thinned down to 1 nm, and more than 90% of the grains have the r-(111) phase structure, and very few m-(-111) and o-(001) grains coexist (Fig. S8(C)). It should be noted that the minor grains categorized as o-(001) for the 1 and 3-nm-thick membranes show almost no periodic modulations in the distance between neighboring Hf/Zr atoms characteristic of the o-(001) structure, in contrast to those in the 5-nm-thick membrane (Figs. 3E-G and S8). The observed cation arrangements imply that structure candidates of the (001)-oriented grains in the 1 and 3-nm-thick membranes are cubic, tetragonal, and rhombohedral. As shown in Fig. S8b, cation arrangements and SAED patterns of these structures projected in the [001] direction are essentially the same and identifying the structural phase of the (001)-oriented grains in the 1 and 3-nm-thick membranes requires further characterizations. **(Main, Page 5, Line 40 to Page 6, Line 1)** The SAED patterns of 1 and 3-nm-thick HZO membranes (Figs. 3(I) and 3(J)) and their analysis based on the SAED simulation (Fig. S9) support that the majority of the grains

in both membranes have the r-(111) structure. Figures 3(K) and S10 plot the SAED intensity against the reciprocal distance determined from the diffraction patterns in Figs. 3(H)-3(J), further validating the results of the phase distribution analysis. The structural models with the bulk (or computed) lattice parameters reproduce the positions of most of the reflections.

ii). Update original Fig. S8-S10 with large area phase distribution, SAED pattern simulation and diffraction intensity simulation of 3-nm-thick hafnia membrane added. The layout of the original figures is also modified. The following descriptions are also added: **(Supplementary Materials Section 5: Page 4 Line 7-18)** ... When the thickness is increased to 3 nm, in addition to a minority of o-(001) grains, an amount of m-(-111) grains slightly increases, while the r-(111) grains still dominate in the membranes, as shown in Fig. S8(D). We note that the minor grains categorized as o-(001) in the 1 and 3-nm-thick membranes show almost no periodic modulation in the distance between neighboring Hf/Zr atoms characteristic of the o-(001) structure, in contrast to those in the 5-nm-thick membrane. The observed cation arrangements imply that structure candidates of the (001)-oriented grains in the 1 and 3-nm-thick membranes are cubic, tetragonal, and rhombohedral. However, as shown in Fig. S8b, cation arrangements of these structures projected in the [001] direction are essentially the same, and identifying the structural phase of the (001)-oriented grains in the 1 and 3-nm-thick membranes requires further characterizations. We note that this sorting of the minor domains in the 1 and 3-nm-thick membranes does not change our main conclusion.

(#1-5) The XRD results presented in Fig. 2(D) and Fig. S7 show a broad band at 22-23° in the 5 nm-thick HZO membrane which cannot be attributed to any hafnia polymorph. The authors should explain its origin. Does it correspond to STO or LSMO, suggesting that the membrane still contains rests of a perovskite oxide and thus is not properly free-standing? Are the results similar for other thickness values?

Author's response:

The broad intensity seen around $2\theta \sim 22^\circ$ is actually background signals coming from a sample holder implemented in our measurement system, which can be confirmed from a diffraction pattern without samples. The diffraction pattern has been included in Fig. S7. We have

also added sentences regarding the origin of the broad intensity at $2\theta \sim 22^\circ$ in the captions of Fig. 2d and S7 to clear up the confusion of readers.

Changes from the original manuscript:

i). Description has been added to the manuscript: **(Supplementary Materials Section 4; Page: 3 Line: 39-40)**...occurred during these periods, which indicates the stability of the metastable phase of HZO. **It should be noted that the board peak in the 2θ region around 22° originates from a sample holder used in our measurement system.**

(#1-6) The identification of the structures of different grains is based on the comparison between the STEM-HAADF experimental images and simulations. Showing other polymorphs and zone axes giving similar images would be useful to reinforce the structural assignment done in the text.

Author's Response:

The supplementary materials (Fig. S8 (C-E) in Section 5) described the phase distribution in 1-, 3- and 5-nm-thick membranes over larger and different areas. In response to the reviewer's comment, we have listed in Fig. S8b the structural models of HZO polymorphs and their FFT patterns so that readers can easily understand our structural assignments in Figs 3, S8 and S9.

Changes from the original manuscript:

i). Update original Fig. S8 by adding the comparison between different polymorphs in Fig. S8(B) and description: **(Supplementary Materials Section 5: Page 4 Line 1-5)** ...rhombohedral structure (r-(111)). **Moreover, this phase identification is further checked by comparing with the FFT patterns simulated from polymorph structures of HZO, and their atomic arrangements, shown in Fig. S8 (B). It can be concluded that the structural phases of grains in our HZO membranes are identified as r-(111), o-(001), o-(111), or m-(-111).**

ii). Add more descriptions about phase distribution: **(Supplementary Materials Section 5; Page 4, Line 5-21)** Our structural phase distribution analysis **over larger observation areas (90 x 90 nm)** indicates that the 1-nm-thick membranes have a majority of r-(111) grains and a minority of o-(001) and m-(-111) grains (Fig. S8(C)). **When the thickness is increased to 3 nm, in addition to a minority of o-(001) grains, an amount of m-(-111) grains slightly increases, while the r-(111) grains still dominate in the membranes, as shown in Fig. S8(D). We note that the minor grains**

categorized as o-(001) in the 1 and 3-nm-thick membranes show almost no periodic modulation in the distance between neighboring Hf/Zr atoms characteristic of the o-(001) structure, in contrast to those in the 5-nm-thick membrane. The observed cation arrangements imply that structure candidates of the (001)-oriented grains in the 1 and 3-nm-thick membranes are cubic, tetragonal, and rhombohedral. However, as shown in Fig. S8b, cation arrangements of these structures projected in the [001] direction are essentially the same, and identifying the structural phase of the (001)-oriented grains in the 1 and 3-nm-thick membranes requires further characterizations. We note that this sorting of the minor domains in the 1 and 3-nm-thick membranes does not change our main conclusion. On the other hand, the 5-nm-thick HZO film is dominated by grains with the o-(111) structure, with a mixture of grains having the o-(001) and m-(111) structures (Fig. S8(E)). The presence of the most stable m-(-111) grains as increasing the thickness implies a thickness-induced structural relaxation.

(#1-7) Strictly speaking, piezoresponse proves the piezoelectric behavior of the films, not necessarily their ferroelectric nature. “Piezoelectricity” could be used instead of “Ferroelectricity” most times in the section beginning in line 191, including its title, and also in Section 6 of the Supplementary Materials.

Author’s Response:

We understand the point raised by the reviewer. We have replaced “ferroelectricity” in Section 6 of Supplementary Materials with “imprint effect”. On the other hand, with “ferroelectricity” used in the sentences describing PFM results in the section beginning from Line 4 on Page 6, we meant the piezoelectric response characteristic of the ferroelectricity. We think that this phrase is understandable through the context in the paragraph. Therefore, we would like to leave “ferroelectricity” as it is. We made some revisions to clarify this point.

Changes from the original manuscript:

i). (Main; Page 3, Line 33-35) To further explore ferroelectricity in the 1-nm-thick o-HZO films, we performed piezoelectric force microscope (PFM) characterizations to measure the piezoelectric response characteristic of the ferroelectricity.

(#1-8) The authors should explain why measuring polarization-field hysteresis curves was not possible in some of the films. For instance, loops similar to Fig. S2 could be shown in Fig. 1 for the 1 nm and 3 nm-thick samples.

Author's Response:

In fact, 1-nm- and 3-nm-thick epitaxial HZO films are too leaky to evaluate polarization values from P-E hysteresis loop measurements. This information has been added in the manuscript. We also presented P-E loops for 1-nm- and 3-nm-thick HZO epitaxial films obtained at the frequency of 100 kHz, and it can be seen that due to the large contribution from leakage current, the ferroelectric polarization cannot be well characterized for these ultrathin films.

Changes from the original manuscript:

i). Incorporated P-E loops of 3-nm- and 1-nm-thick thin films into Fig. S2 with further descriptions: **(Supplementary Materials Section 2; Page 2 Line 38-44) The switching polarization ($2P_r$), determined based on the assumption that the leakage currents equally contribute to the charges measured under the P(N) and U(D) voltage pulses in PUND measurements, also keeps increasing with electric fields (Fig. S2(D)), indicating that leakage currents in the 1 and 3-nm-thick epitaxial films are very large and that eliminating the leakage current contribution is difficult even for PUND measurements.**

(#1-9a) I would prefer the term “crystal structure” instead of “atomic structure” in lines 154, 156 and following.

Author's Response:

As suggested by the reviewer, “atomic structure” in Line 42 and Line 43 on Page 4, and in some other lines have been changed to “crystal structure”.

(#1-9b) The crystal directions and orientations should be indicated as [hkl] instead of (hkl). The XRD reflections should be indicated as hkl instead of (hkl).

Author's Response:

We understand the crystallographic expression rules pointed out by the reviewer. However, there is the convention that the films' orientation is indicated with the (hkl) notation, and

diffraction peak indices from film samples are shown as (hkl). We note that nature family journals accept this convention of crystallographic expression for thin film samples (for example, *Nature Mater* 2018, 17, 1095.). In our manuscript, we would like to follow the convention.

(#1-9c) “Poling” should be used instead of “polling”.

Author’s Response:

The typo has been corrected.

(#1-9d) The paper with DOI 10.1021/cm021111v is an important one in the field of epitaxial stabilization of oxide films. It can be added together with Refs. 9-11.

Author’s Response:

The reference suggested by the reviewer has been added in the revised version as Ref. 12.

(#1-9e) The length of the scale bars should be clarified in all the figures, at least Figs. 2C, S3, S6A and S11A.

Author’s Response:

The length of the scale bars has been added in the figure captions.

(#1-9f) (E) label is missing in the caption of Fig. 4.

Author’s Response:

The second label (D) in the figure caption should be (E) and has been corrected in the revised version.

(#1-9g) Fig. S9(C) is mentioned instead of Fig. S12(C) in the last paragraph of section 7 of the Supplementary Materials.

Author’s Response:

As pointed out by the reviewer, “Fig. S9(C)” in the last paragraph of section 7 of the Supplementary Materials should be “Fig. S12(C)”. We have corrected this mistake in the revised version.

(#1-9h) Ref. 2 in the caption of Table S1 is likely Ref. 1.

Author’s Response:

Ref. 2 in the caption of Table S1 was corrected to Ref. 1.

(#1-9i) “Topography” should be used instead of “Topology” in Fig. S12(A).

Author’s Response:

“Topology” in Fig. S12 has been changed to “Topography”.

Comments from Reviewer #2 (Remarks to the Author):

(#2-0) The authors reported ferroelectric freestanding hafnia membranes with metastable rhombohedral structure down to 1 nm. The result is interesting. However, the reviewer believes that accurately evaluating electrical and physical properties of 1 nm-thick films must be more accurately evaluated with the state-of-the-art techniques. The reviewer can agree that it is highly challenging, but it is need for the publication of the work in highly esteemed journal such as Nature Communications. Furthermore, some discussions were not clear for the reviewer considering the recent understanding of ferroelectric HZO films. Therefore, the reviewer believed that this work can be reconsidered after a mandatory major revision. The detailed comments are attached below.

Author's Response:

We also thank reviewer#2 for offering many suggestions to improve our work, which is now more helpful for the broader audience to understand. As shown below, we have made a great effort to answer these questions with revisions colored in red. Thus, we hope our revised work can now be recommended for publication in *Nature Communications*.

(#2-1) In the title, what is the meaning of monolayer? 1 nm is already 2 unit cells thick with containing four metal layers and four oxygen layers. Clarify this point.

Author's Response:

As pointed out by the reviewer, we admit that the term “monolayer” is confusing. We have removed “monolayer” from the manuscript. We also have changed the title to “Ferroelectric freestanding hafnia membranes with metastable rhombohedral structure down to 1-nm-thick”.

(#2-2) What is the rhombohedral space group considered in this work? There have been reports on epitaxial HZO films epitaxially grown using PLD. However, there is a recent report on metal-rich rhombohedral phase published in Science. Please clarify this point.

Author's Response:

Because obtaining oxygen atomic positions in our membranes and determining their structural space groups are difficult, the space group of the rhombohedral phase is assumed to be

$R3m$ (No.160) in our work. This rhombohedral space group is the same as that reported previously (*Science* 2023, 381, 558. and *Nature Mater* 2018, 17, 1095.). The details of our structural models used in this work are provided in Table. S1.

(#2-3) The authors wrote that “Furthermore, their reflection positions in 2θ shift toward the lower 2θ side with decreasing thickness, revealing large lattice expansions along the out-of-plane [111] direction in the monolayer limit.” However, what is the evidence for this statement? The reviewer believes that a further analysis evaluating both out-of-plane and in-plane lattice parameters is needed.

Author’s Response:

The original sentence was unclear and misleading. With the sentence, we meant that the interplanar distance of the HZO epitaxial films elongates with decreasing thickness. We have made revisions to the sentence.

Changes from the original manuscript:

i). Rephrase has been made to the manuscript: **(Main; Page 3 Line 10-12)** Furthermore, the (111) HZO reflection positions shift toward the lower 2θ side with decreasing thickness, **indicating that the interplanar distance along the out-of-plane [111] direction elongates with decreasing thickness.**

(#2-4) The switching field in figure 1c seems extremely high especially for 1 and 3 nm-thick film. It is reaching the field required for the electron tunneling already. Can we believe that the measured switching polarization is free from the charge injection and tunneling? The reviewer has the same concern on figure S11.

Author’s response:

We agree with the reviewer that leakage currents, such as charge injections and tunneling currents, possibly influence the polarization measured for our ultrathin HZO films. As described in section 8 in the Supplementary Materials, for PUND measurements, the polarization values are calculated by assuming that the leakage currents are equally contributed to the charges measured under the P(N) and U(D) voltage pulses. However, this assumption might not be true when leakage

currents are large, as in cases of ultrathin films. Therefore, to eliminate these non-ferroelectric (leakage current) contributions from the measured polarization more precisely, we estimated the non-ferroelectric contribution from charges profiles measured with the N and D voltage pulses and determined the switching polarization (or $2P_r$). The details about this approach are added to the manuscript (Supplementary Materials Section 8, Page 6 Line 4-23). The typical results of this analysis for the membranes investigated in this study are summarized in Table. R1 (Labeled as Table. S2 in Supplementary Materials) as shown below. It can be seen that the polarization values that non-ferroelectric currents might induce are estimated to be at most $6 \mu\text{C}/\text{cm}^2$, which is much smaller than the observed $2P_r$ (around $30 \mu\text{C}/\text{cm}^2$ for the membrane samples before the correction), indicating that the leakage current (non-ferroelectric) contribution to the observed polarization is less significant. The case for epitaxial thin films shares the same situation. The PUND results in Figs. 1(C) and 4(E) are also corrected based on these analyses and the error bars, which indicate variations in the polarization values obtained by ten repeated measurements, are also added. The corrected $2P_r$ values for the transferred membranes are all around $25 \mu\text{C}/\text{cm}^2$.

Thickness	I_{leakage} (A)		$\Delta Q_{\text{leakage}}$ ($\mu\text{C}/\text{cm}^2$)	$2P_{r\text{-Corrected}} (\mu\text{C}/\text{cm}^2)$ $= 2P_{r\text{-PUND}} - \Delta Q_{\text{leakage}}$
1 nm	$-1.14\text{e-}07 \pm 1.35\text{e-}09$ (N)	$-1.06\text{e-}07 \pm 1.17\text{e-}09$ (D)	4.9	24.7
	$1.69\text{e-}07 \pm 1.02\text{e-}09$ (P)	$1.64\text{e-}07 \pm 1.16\text{e-}09$ (U)	2.7	27.6
3 nm	$-1.97\text{e-}07 \pm 1.81\text{e-}09$ (N)	$-1.73\text{e-}07 \pm 1.17\text{e-}09$ (D)	4.7	27.7
	$1.48\text{e-}07 \pm 1.86\text{e-}09$ (P)	$1.33\text{e-}07 \pm 1.20\text{e-}09$ (U)	3.0	30.4
5 nm	$-9.08\text{e-}08 \pm 8.7\text{e-}10$ (N)	$-8.16\text{e-}08 \pm 5.85\text{e-}10$ (D)	5.8	24.7
	$9.07\text{e-}08 \pm 1.55\text{e-}09$ (P)	$8.03\text{e-}08 \pm 5.18\text{e-}10$ (U)	6.5	24.9

Table. R1. Non-ferroelectric contribution extracted from the charge profiles in the PUND measurements (**Labeled as Table. S2 in Supplementary Materials**).

In addition, the $2P_r$ values we obtained from PUND are reproducible. As shown in Fig. R5 (Labeled as Fig. S13(G) in Supplementary Materials), the $2P_r$ values remain almost unchanged against ten cycles of PUND voltage application. We also note that essentially the same PUND results for the 1 nm-thick membranes were obtained from different junctions (Fig. S14 in Section 8 in the Supplementary Materials). It should also be pointed out that when the leakage currents

largely contribute to $2P_r$ from PUND measurements, the $2P_r$ value would keep increasing with the electric field and would not saturate as seen in the cases of the 1- and 3-nm-thick HZO epitaxial films (Fig. S2(D)). On the other hand, as shown in Fig. 4(E), $2P_r$ for the HZO membranes are well saturated against the applied field.

Fig. R5. Switchable polarizations for the 1-, 3-, and 5-nm-thick HZO membranes, obtained from ten cycles of PUND measurements (Labeled as Fig. S13(G) in Supplementary Materials).

All our observations point to the conclusion that the leakage contribution to the polarization is secondary, although the switching field is large for the ultrathin membranes. Large switching fields comparable to those for our samples were also observed in previous reports on ultrathin hafnia films (*Nature*, 2020, 580, 478.).

For our PFM characterization, we measured the out-of-plane piezoelectric response under zero external d.c. bias so as to minimize the contribution of charge injections and tunneling currents to the measured piezoelectric response. As depicted in Fig. S11(B) in Supplementary Materials, external d.c. biases were used only for switching polarizations, and the influence of d.c.-bias-induced tunneling currents would be gone when measuring the piezoelectric response under zero d.c. bias. We also note that the hysteresis behavior of off-field piezoresponse for HZO membranes remains almost unchanged against the voltage sweep direction and measurement repetitions (Fig. S11 and Section 6 in Supplementary Materials). If charge injections and tunneling currents contribute to the measured piezoelectric response dominantly, as pointed out by the reviewer, we expect to see some changes in the hysteresis loops depending on the voltage sweep

direction and measurement repetitions. However, such changes are not seen in our PFM results, implying that the influence of charge injections and tunneling currents on piezoelectric response is secondary.

All these discussions have been included in the manuscript.

Changes from the original manuscript:

i). Table. R1 has been added to the manuscript (**Labeled as Table. S2 in Supplementary Materials**), and descriptions about how we extracted non-ferroelectric contribution are added to the manuscript: (**Supplementary Materials Section 8, Page 6 Line 4-23**) We note that the polarization calculation described above assumes that the leakage currents equally contribute to the charges measured under the P(N) and U(D) voltage pulses. However, this assumption might not be correct when leakage currents are large. Therefore, to eliminate these non-ferroelectric (leakage current) contributions and determine polarization values from PUND measurements more precisely, we estimated the non-ferroelectric contribution from charge profiles measured with the N and D voltage pulses, and corrected the $2P_r$ values that were calculated as $2P_r = [(Q_2 - Q_1) - (Q_3 - Q_2)]$. The charges originating from leakage current flow while rectangular electrical pulse was applied ($\Delta t = 25 \mu s$) can be calculated as $Q_{leakage} = \int I_{leakage} dt$, in which $I_{leakage}$ should be constant during the voltage pulse application and can be obtained as the slope of charge profiles (indicated with purple dot lines in the right panel of Fig. S13(A)). To extract $I_{leakage}$ associated with each voltage pulse application (P, U, N, and D pulses), the charge profiles in the time region between 15 and 25 μs after each voltage pulse was applied were linear-fitted and $Q_{leakage}$ was calculated as $Q_{leakage} = |I_{leakage}| \cdot \Delta t$ where $\Delta t = 25 \mu s$. The results are summarized in Table. S2. The difference in the leakage current contribution between the P and U (or N and D) pulses are thus estimated as $\Delta Q_{leakage} = (|I_{leakage-P}| - |I_{leakage-U}|) \cdot \Delta t$ (or $= (|I_{leakage-N}| - |I_{leakage-D}|) \cdot \Delta t$). If the leakage current would contribute to $2P_r$ largely, we should see a significant amount of $\Delta Q_{leakage}$. However, $\Delta Q_{leakage}$ in our measurements is estimated to be at most $6 \mu C/cm^2$, which is much smaller than the observed $2P_r$ (around $30 \mu C/cm^2$ before the correction), indicating that the leakage current contribution is less significant. The $2P_r$ values plotted in the revised version, except for the ones in Fig. S2(D), are all corrected for the leakage current contribution based on this approach.

ii). Update Fig. S13 by adding Fig. R6 (labeled as Fig. S13(G)) and adding descriptions: (**Supplementary Materials Section 8, Page 6 Line 24-27**) ...The charge profiles during PUND measurements are consistent over repeated measurements (Fig. S13(E-F)), and the $2P_r$ values are

estimated to be steady against the measurement cycles (Fig. S13(G)), further validating that the non-ferroelectric contribution in our transferred membranes is less important.

iii). Update Fig. 1(C) with PUND results of 3-nm- and 1-nm-thick HZO epitaxial films moved to Fig. S2(D) with additional descriptions in the manuscript: **(Supplementary Materials Section 2, Page 2 Line 38-43)** The switching polarization ($2P_r$), determined based on the assumption that the leakage currents equally contribute to the charges measured under the P(N) and U(D) voltage pulses in PUND measurements, also keeps increasing with electric fields (Fig. S2(D)), indicating that leakage currents in the 1 and 3-nm-thick epitaxial films are very large and that eliminating the leakage current contribution is difficult even for PUND measurements.

iV). Correct the PUND results and update the figures (Fig. 1(C) and Fig. 4(E)) with error bars added to the extracted $2P_r$ values.

(#2-5) It was not clear for the reviewer whether the HZO films were completely crystallized from figure 2b. It is well known that the crystallization of HZO films thinner than 3 nm. Please provide more analysis or explanation on the crystallization degree of the HZO films.

Author's Response:

To obtain information about crystallinity of the HZO epitaxial films, we performed omega-scans of the (111) HZO reflections for 1-, 3-, and 5-nm-thick HZO films. The results are provided in Fig. R6 (Labeled as Fig. S1(C) in Supplementary Materials). The half-width full maximum (HWFHM) of the (111) reflections for all the samples is as low as 0.05° , ensuring the crystallinity of our HZO epitaxial films. In addition, we evaluated the crystallinity of HZO membranes from plane-view HAADF-STEM images provided in Fig. S8(C-E), as shown in Fig. R7 (Labeled as Fig. S8(C) in Supplementary Materials). A small amount ($\sim 4\%$ at most) of amorphous grains are detected in the 1-nm-thick membranes, while no amorphous grains are found for the 3- and 5-nm-thick membranes. These observations confirm the crystallinity of the HZO membranes. These pieces of information have been included in the manuscript.

Fig. R6 Omega-scan for the 1-, 3-, and 5-nm-thick HZO epitaxial films. (Labeled as Fig. S1(C) in Supplementary Materials).

Fig. R7 HAADF-STEM images of the 1-nm-thick HZO membrane with amorphous region colored in pink. (Labeled as Fig. S8(C) in Supplementary Materials).

Changes from the original manuscript:

i). Incorporated Fig. R6 into Fig. S1 with descriptions: (Main; Page 3 Line 16-19): and the half-width full maximum (HWFm) of the (111) reflections for all epitaxial films studied here, which were evaluated by performing omega scans, are as low as 0.05° (Fig. S1), ensuring the crystallinity of our HZO epitaxial films. (Supplementary Materials Section 1; Page 2 Line 13-

16): Furthermore, the crystallinity of the epitaxial thin films was evaluated from omega scans of the (111) HZO reflections. The results are shown in Fig. S1(C). It can be seen that the half-width full maximum (HWFMM) of the (111) reflections for all epitaxial films studied here are as low as 0.05° , ensuring the crystallinity of our HZO epitaxial films.

ii). Incorporated Fig. R7 into Fig. S8 with descriptions: **(Main; Page 5 Line 5-8):** We also note that while a tiny amount of amorphous grains ($< 4\%$) are detected in the 1-nm-thick membranes (as indicated in Fig. S8(C)), no amorphous grains are observed for 3- and 5-nm-thick membranes. These observations ensure the crystallinity of the HZO membranes. **(Supplementary Materials Section 5; Page 4 Line 24-26):** We also note that while a tiny amount of amorphous grains ($< 4\%$) are detected in 1-nm-thick hafnia membranes (indicated in purple in Fig. S8(C)), no amorphous grains are observed for 3-nm- and 5-nm-thick membranes. These observations ensure the crystallinity of the HZO membranes.

(#2-6) The spatial phase map and selective area electron diffraction in figures S8 and S9 is very interesting. However, what is the accuracy of this analysis? Provide more details of the analysis. Which phases are the most difficult phase to distinguish from each other? How accurately does the phase analysis differentiate the relative position of oxygen ions which should be the key to differentiate the different crystallographic structure of HZO films? It was already very tough to evaluate the 10 nm-thick films using these techniques. How did the authors improve this analysis applicable to 1 nm membrane? Please compare the analysis in this work to that in previous works.

Author's Response:

To accurately determine the structural phase from HAADF-STEM images in Fig.S8, we analyzed high-resolution images with reduced image drift and improved signal-to-noise ratio, which were obtained by acquiring 30 images and combining them after their drift correction. The structural phase was determined by comparing cation arrangements seen in HAADF-STEM images with those expected from the structural models shown in Fig. S8(A).

We also note that the SAED analysis (Fig. S9) allows identification of the structural phase. To ensure the accuracy of electron diffraction images that we obtained from membrane samples, the diffraction images were calibrated using a single crystal of SrTiO_3 with lattice parameter

precisely determined by XRD ($a = 0.3905$ nm, *Phys. Rev. B* **98**, 134114 (2018)). However, even with calibrated SAED patterns, distinguishing between the orthorhombic and monoclinic phases might be difficult because some diffraction spots from these phases, for example, Orthorhombic 202 and Monoclinic 022 spots, appear at almost the same position. Nonetheless, the cation arrangements of these structural phases totally differ, and these phases are distinguishable from the high-resolution HAADF-STEM images. We think that our phase identification (phase separation) is correct because structural phases are identified by combining the analysis of STEM images and SAED patterns. In the revised version, we have added these pieces of information to the Method section (Main, Method; Page 9 Line 20-21). We also found from the careful structural analysis that the minor grains categorized as o-(001) for the 1 and 3-nm-thick membranes show almost no periodic modulations in the distance between neighboring Hf/Zr atoms characteristic of the o-(001) structure, in contrast to those in the 5-nm-thick membrane (Figs. 3E-G and S8). The observed cation arrangements imply that structure candidates of the (001)-oriented grains in the 1 and 3-nm-thick membranes are cubic, tetragonal, and rhombohedral. As shown in Fig. S8b, cation arrangements and SAED patterns of these structures projected in the [001] direction are essentially the same and identifying the structural phase of the (001)-oriented grains in the 1 and 3-nm-thick membranes requires further characterizations. In the revised manuscript, we keep sorting this type of minority grain as o-(001) for simplicity, and this sorting does not change our main conclusion. Details of STEM observations and structural phase analysis have been included in the Supplementary Materials. The manuscript has also been revised accordingly (Main, Page 5, Line 27-34, Supplementary Materials Section 5: Page 4 Line 9-18, and the figure captions of Figs. 3, S8 and S9).

Although it is challenging and difficult to determine the oxygen atomic position precisely, it is still possible to identify the structural phases from plane-view HAADF-STEM images for HZO membranes because the cation configuration of HZO differs depending on its structural phases, as shown in the previous report (*Adv Mater* **34**, e2109889 (2022)). Although the previous report showed that the structural phase of HZO could be identified only from STEM images, we further improved structural phase identification based on both STEM images and SAED patterns.

We would also like to point out that the plane-view HAADF-STEM observation technique has not been utilized for characterizing hafnia thin films since hafnia-based thin films are always

heterostructured with electrodes' materials, blocking the hafnia thin films from being evaluated in plane-view due to the overlapping of atoms of hafnia and electrodes along an observing direction. Currently, cross-sectional HAADF-STEM observation is widely used for characterizing hafnia-based thin films. However, the coexistence of multiple in-plane domains whose sizes are extremely small (especially when scaling down) makes observing the hafnia thin films on the atomic level in cross-sectional directions very tough and luck-dependent. On the other hand, the freestanding membranes enable plane-view observation with clearly distinctive cation arrangements in different structural phases without the influences of electrodes' materials. Furthermore, to obtain high-resolution HAADF-STEM images, we adjust and minimize the image drift during the image acquisition and finely calibrate other external factors, such as magnetic field lens and focus drift.

Changes from the original manuscript:

i). Add descriptions to the manuscript: **(Main, Method; Page 9 Line 20-21):** ... their intensity profiles **were calibrated using a single crystal of SrTiO₃ with lattice parameter precisely determined by XRD and** simulated with the Single Crystal and CrystalDiffract software (CrystalMaker Software Limited).

(#2-7) The result in figure S10 was difficult to understand. Why is there no consideration of orthorhombic Pca21 in red color for the result of 1 nm-thick film in the bottom panel?

Author's Response:

As we described in our response to the comment (#2-6), we identified the structural phases based on HAADF-STEM images and SAED pattern. As pointed out by the reviewer, the reciprocal distance dependence of diffraction intensities in 1-nm-thick membranes (Fig. S10) is indeed similar to that of the orthorhombic phase. Nevertheless, the cation arrangement seen in the HAADF images (Fig. 4(A-D) and Fig. S8(A)) matches well with the one expected from the rhombohedral phase and totally differs from the one from the orthorhombic phase. Thus, we did not include the orthorhombic phase when analyzing the reciprocal distance dependence of diffraction intensities in 1-nm-thick membranes. We made additional descriptions to the manuscript to clear the confusion.

Changes from the original manuscript:

i) Add descriptions to the manuscript: **(Supplementary Materials Section 5; Page 4 Line 39-44)** ... with the rhombohedral and orthorhombic structures. It should be noted that distinguishing between the orthorhombic and monoclinic phases might be difficult because some diffraction spots from these phases, for example, Orthorhombic 202 and Monoclinic 022 spots, appear at almost the same position. Similarly, distinguishing between Orthorhombic 022 and Rhombohedral 220 is also difficult. Nonetheless, the cation arrangements of these structural phases totally differ, and these phases are distinguishable from the high-resolution HAADF-STEM images.

Comments from Reviewer #3 (Remarks to the Author)

(#3-0) This work reveals the ferroelectric properties of HZO membranes down to atomic thickness. It also points out the transformation from rhombohedral phase to orthorhombic phase when the film thickness increase. Overall the result is very useful for the development of 2D ferroelectrics. My main concern is regarding the ferroelectric properties extracted from PUND tests.

Author's response:

We appreciate the reviewer for stating our work as “very useful for the development of 2D ferroelectrics”. We also thank reviewer#3 for helping us to improve PUND analysis for more reliable results. After revising the manuscript (revisions colored in red) based on these suggestions, we hope our work can now be recommended for publication in *Nature Communications*.

(#3-1) There is only one P-E loop demonstrated, in Figure S2 of the SI, but no such result shown in the main text. Even though this figure is related to the thick 5 nm case, the loop is not well saturated. Given that the authors claim ferroelectricity from 1 nm to 5 nm, then P-E loop for each thickness should be provided.

Author's Response:

The 1-nm- and 3-nm-thick epitaxial HZO films are too leaky to evaluate polarization values from P-E hysteresis loop measurements. This information has been added to the revised version of the manuscript. We presented P-E loops for 1-nm- and 3-nm-thick HZO epitaxial films obtained at the measurement frequency of 100 kHz in Fig. S2(B-C), and the obtained loops are found to be contaminated with leakage current and it can be seen that due to large contribution from leakage current, the ferroelectric polarization cannot be well characterized for these ultrathin films.

Changes from the original manuscript:

i). Incorporated P-E loops of 3-nm- and 1-nm-thick thin films into Fig. S2 with further descriptions: **(Supplementary Materials; Page 2 Line 36-43) On the other hand, the 1 and 3-nm-thick epitaxial films are too leaky to evaluate switching polarization from P-E loop and PUND measurements as shown in Figures S2 (B-D), and it can be seen that the P-E loops were heavily contaminated with leakage currents. The switching polarization ($2P_r$), determined based on the**

assumption that the leakage currents equally contribute to the charges measured under the P(N) and U(D) voltage pulses in PUND measurements, also keeps increasing with electric fields (Fig. S2(D)), indicating that leakage currents in the 1 and 3-nm-thick epitaxial films are very large and that eliminating the leakage current contribution is difficult even for PUND measurements.

(#3-2) The 2Pr values extracted from PUND tests are questionable. From Figure S2 it is shown that the loop is not saturated, with giant possible leakage current. It is well-known that leakage current can be large in ultra-thin films. On the other hand, the PUND result should eliminate the influence from these leakage. However, the reported 2Pr value from PUND in the main text exactly correspond to the 2Pr value extracted from the leakage-containing P-E loops, and without considering the retention loss (there is a considerable gap in the P-E loop, indicating retention problem). Hence, the argument for such high 2Pr values from PUND is questionable. A reliable 2Pr estimation must be given based correctly on PUND results.

Author's Response:

To minimize the leakage contribution and evaluate 2Pr from P-E hysteresis loops, we obtained P-E loops at high frequencies to suppress the leakage/breakdown while better polarize samples at the same time, the frequency is up to 100 kHz and is the highest frequency in our instrument. The results for the 5nm-thick HZO epitaxial films have been shown in Fig. R8 (Labeled as Fig. S2(A) in Supplementary Materials). The 2Pr values obtained from the 10 kHz and 100 kHz loops are $24 \mu\text{C}/\text{cm}^2$ and $20 \mu\text{C}/\text{cm}^2$, respectively, and these 2Pr values are comparable to 2Pr value ($22 \mu\text{C}/\text{cm}^2$ before the correction of the leakage current contribution) obtained from PUND measurements. These results indicate that the leakage-current contributions on the PUND measurements for the 5-nm-thick HZO epitaxial films are not significant, and the 2Pr value evaluated from PUND measurements is reasonable. We also note that, as shown in Fig. 3, the 5-nm-thick-films consist of grains with the ferroelectric orthorhombic phase and the paraelectric monoclinic phase, which might be a reason why the loop is not saturated.

Fig. R8. P-E loops and I-E loops of 5-nm-thick epitaxial thin films measured at 1 kHz, 10 kHz, and 100 kHz. (Labeled as Fig. S2(A) in Supplementary Materials).

To further improve our analysis of PUND results for the 5-nm-thick epitaxial thin film, we extracted leakage current (non-ferroelectric) contribution in our 2Pr values obtained from PUND measurements. The details of the approach are added to the manuscript (Supplementary Materials; Page 5 Line 37 to Page 6 Line 10). The extracted non-ferroelectric contribution in the epitaxial sample is only $2.4 \mu\text{C}/\text{cm}^2$, which is much smaller than the 2Pr values evaluated from PUND measurements, indicating that non-ferroelectric contribution is less important in our PUND measurements. The PUND results are corrected based on this non-ferroelectric contribution analysis. The same correction is also performed for HZO membranes whose leakage contribution is summarized in Table R1 (Labeled as Table. S2 in Supplementary Materials), and some more details of our analysis can also be found in our response to the comment (#2-4).

	I_{leakage} (A)	$\Delta Q_{\text{leakage}}$ ($\mu\text{C}/\text{cm}^2$)	$2P_{\text{r-Corrected}} (\mu\text{C}/\text{cm}^2)$ $= 2P_{\text{r-PUND}} - \Delta Q_{\text{leakage}}$
1 nm			
-1.14e-07 ± 1.35e-09 (N)	-1.06e-07 ± 1.17e-09 (D)	4.9	24.7
1.69e-07 ± 1.02e-09 (P)	1.64e-07 ± 1.16e-09 (U)	2.7	27.6
3 nm			
-1.97e-07 ± 1.81e-09 (N)	-1.73e-07 ± 1.17e-09 (D)	4.7	27.7
1.48e-07 ± 1.86e-09 (P)	1.33e-07 ± 1.20e-09 (U)	3.0	30.4
5 nm			
-9.08e-08 ± 8.7e-10 (N)	-8.16e-08 ± 5.85e-10 (D)	5.8	24.7
9.07e-08 ± 1.55e-09 (P)	8.03e-08 ± 5.18e-10 (U)	6.5	24.9

Table. R1. Non-ferroelectric contribution extracted from the charge profiles in the PUND measurements (Labeled as Table. S2 in Supplementary Materials)

Changes from the original manuscript:

i). Revise PUND measurements in Fig. 1(C) by correcting $2P_r$ values for the 5-nm-thick epitaxial films by subtracting the non-ferroelectric contribution.

ii). Update Fig. 1(C) with PUND results of 3-nm- and 1-nm-thick HZO thin films moved to Fig. S2(D) and with additional descriptions in the manuscript: **(Supplementary Materials Section 2, Page 2 Line 38-43)** The switching polarization ($2P_r$), determined based on the assumption that the leakage currents equally contribute to the charges measured under the P(N) and U(D) voltage pulses in PUND measurements, also keeps increasing with electric fields (Fig. S2(D)), indicating that leakage currents in the 1 and 3-nm-thick epitaxial films are very large and that eliminating the leakage current contribution is difficult even for PUND measurements.

iii). Incorporated Fig. R8 into Fig. S2 with additional descriptions: **(Supplementary Materials Section 2; Page 2 Line 23-30)** We note that as the measurement frequency is increased, the leakage current contribution is further reduced. The $2P_r$ obtained from the P-E loops measured at 100 kHz is $20 \mu\text{C}/\text{cm}^2$, which is almost the same as $2P_r$ obtained from the 1 kHz and 10 kHz loops ($24 \mu\text{C}/\text{cm}^2$). Furthermore, these $2P_r$ values are slightly larger than the ones obtained from PUND measurements ($19 \mu\text{C}/\text{cm}^2$, Fig.1(C)). These results indicate that the leakage current contribution in P-E loops is less significant and that the non-saturated P-E loops probably result from paraelectric components co-existing in the films, for example, grains with the monoclinic structure.

iv). Add descriptions to the manuscript about the leakage current contribution analysis: **(Supplementary Materials Section 8, Page 6 Line 4-23)** We note that the polarization calculation described above assumes that the leakage currents equally contribute to the charges measured under the P(N) and U(D) voltage pulses. However, this assumption might not be correct when leakage currents are large. Therefore, to eliminate these non-ferroelectric (leakage current) contributions and determine polarization values from PUND measurements more precisely, we estimated the non-ferroelectric contribution from charge profiles measured with the N and D voltage pulses, and corrected the $2P_r$ values that were calculated as $2P_r = [(Q_2 - Q_1) - (Q_3 - Q_2)]$. The charges originating from leakage current flow while rectangular electrical pulse was applied ($\Delta t = 25 \mu\text{s}$) can be calculated as $Q_{\text{leakage}} = \int I_{\text{leakage}} dt$, in which I_{leakage} should be constant during the voltage pulse application and can be obtained as the slope of charge profiles (indicated with purple dot lines in the right panel of Fig. S13(A)). To extract I_{leakage} associated with each voltage pulse

application (P, U, N, and D pulses), the charge profiles in the time region between 15 and 25 μs after each voltage pulse was applied were linear-fitted and Q_{leakage} was calculated as $Q_{\text{leakage}} = |I_{\text{leakage}}| \cdot \Delta t$ where $\Delta t = 25 \mu\text{s}$. The results are summarized in Table. S2. The difference in the leakage current contribution between the P and U (or N and D) pulses are thus estimated as $\Delta Q_{\text{leakage}} = (|I_{\text{leakage-P}}| - |I_{\text{leakage-U}}|) \cdot \Delta t$ (or $= (|I_{\text{leakage-N}}| - |I_{\text{leakage-D}}|) \cdot \Delta t$). If the leakage current would contribute to 2Pr largely, we should see a significant amount of $\Delta Q_{\text{leakage}}$. However, $\Delta Q_{\text{leakage}}$ in our measurements is estimated to be at most $6 \mu\text{C}/\text{cm}^2$, which is much smaller than the observed 2Pr (around $30 \mu\text{C}/\text{cm}^2$ before the correction), indicating that the leakage current contribution is less significant. The 2Pr values plotted in the revised version, except for the ones in Fig. S2(D), are all corrected for the leakage current contribution based on this approach.

REVIEWERS' COMMENTS

Reviewer #1 (Remarks to the Author):

The authors have addressed all my comments to the previous version in a thorough, satisfactory way. I thus recommend publication of the paper in Nature Communications.

I just suggest these minor changes and one typo:

Fig. S1(A): please indicate that the intensity scale in the XRR plot is logarithmic.

Fig. S1(C): please indicate the 2theta value for each rocking curve.

Supplement, line 106: "board" should be "broad".

Reviewer #2 (Remarks to the Author):

The reviewer appreciates the authors' efforts. The manuscript is ready for publication.

Reviewer #3 (Remarks to the Author):

My concerns regarding this paper were mainly related to the P-E loops and the extraction of polarization parameters without unexpected influence from leakage. In the revised manuscript, the authors have provided hysteresis loops from even thinner films, and the polarization accuracy problem has been resolved. The work now is acceptable.

Comments from the reviewer#1

(#1-1) Fig. S1(A): please indicate that the intensity scale in the XRR plot is logarithmic.

Author's response:

The label of the vertical axis of Figure S1(A) has been changed to "Log Intensity (arb. units)".

(#1-2) Fig. S1(C): please indicate the 2theta value for each rocking curve.

Author's response:

The 2θ values for each rocking curve have been added in the caption of Fig. S1(C).

(#1-3) Supplement, line 106: "board" should be "broad".

Author's response:

The typo indicated by the reviewer has been corrected.